# Dynamic nucleosome landscape elicits a noncanonical GATA2 pioneer model

Tianbao Li[1,4], Qi Liu[1,4], Zhong Chen [2,4], Kun Fang [1], Furong Huang[2], Xueqi Fu[3], Qianben Wang [2,5✉] & Victor X. Jin [1,5✉]

Knowledge gaps remain on how nucleosome organization and dynamic reorganization are governed by specific pioneer factors in a genome-wide manner. In this study, we generate over three billons of multi-omics sequencing data to exploit dynamic nucleosome landscape governed by pioneer factors (PFs), FOXA1 and GATA2. We quantitatively define nine functional nucleosome states each with specific characteristic nucleosome footprints in LNCaP prostate cancer cells. Interestingly, we observe dynamic switches among nucleosome states upon androgen stimulation, accompanied by distinct differential (gained or lost) binding of FOXA1, GATA2, H1 as well as many other coregulators. Intriguingly, we reveal a noncanonical pioneer model of GATA2 that it initially functions as a PF binding at the edge of a nucleosome in an inaccessible crowding array. Upon androgen stimulation, GATA2 re-configures an inaccessible to accessible nucleosome state and subsequently acts as a master transcription factor either directly or recruits signaling specific transcription factors to enhance WNT signaling in an androgen receptor (AR)-independent manner. Our data elicit a pioneer and master dual role of GATA2 in mediating nucleosome dynamics and enhancing downstream signaling pathways. Our work offers structural and mechanistic insight into the dynamics of pioneer factors governing nucleosome reorganization.

[1] Department of Molecular Medicine, University of Texas Health Science Center at San Antonio, San Antonio, TX 78229, USA. [2] Department of Pathology and Duke Cancer Institute, Duke University School of Medicine, Durham, NC 27710, USA. [3] Edmond H. Fischer Signal Transduction Laboratory, College of Life Sciences, Jilin University, Changchun 130012, China. [4]These authors contributed equally: Tianbao Li, Qi Liu, Zhong Chen. [5]These authors jointly supervised this work: Qianben Wang, Victor X. Jin. ✉email: qianben.wang@duke.edu; jinv@uthscsa.edu

Nucleosome organization (positioning, spacing, and regularity) plays a central role in gene regulation[1,2]. The dynamic nucleosome reorganization is the interplay among nucleosome, wrapped DNA, and nucleosome-binding factors such that nucleosomes sterically occlude their wrapped DNA from DNA-binding factors and ATP-dependent chromatin remodelers unwrap nucleosomal DNA or slide nucleosomes to reposition along DNA[3,4]. Recent genome-wide nucleosome mapping highlighted functionally important regular nucleosomal arrays as their array regularity is often aligned at biological features[5,6]. In contrast, impaired genic arrays are correlated with increased cryptic transcription with suppressive activities[7]. High-resolution mapping studies suggested that DNA contains information specifying the position of nucleosomes called positioning code which facilitates the shunting of nucleosomes from one array to another by chromatin remodeling machines[8–10]. Some studies also showed nucleosomes are subject to extensive reversible post-translational modifications that can alter the local chromatin structure poised for activation and transcription[11,12]. Other studies further elucidated coordinated and antagonistic functional relationships between nucleosome remodeling and modifying machineries[13–15].

A group of special transcription factors (TFs) called pioneer factors (PFs) such as FOXA families[16,17], GATA families[18–20], PAX7[15], HOXB13[21], and P53[22] can access target DNA sequences on nucleosomes. A paradigm of competence for transcription is thus established that nucleosome-binding properties of PFs can engage in the assembly of regulatory factors on the DNA by either opening the chromatin locally, positioning nucleosomes, or enabling intrinsic cooperative binding effects among other TFs[23,24], or directly recruiting other chromatin modifiers and coregulators[25]. Despite many advances in the understanding of nucleosome dynamics, there still remain knowledge gaps on how nucleosome organization and dynamic reorganization are governed by specific pioneer factors in a genome-wide manner.

Many studies[26–28], including ours[19,29], have found both FOXA1 and GATA2 act as PFs to trigger the androgen-induced androgen receptor (AR) signaling pathway. It is believed that both PFs can recruit chromatin modifiers, chromatin remodelers, and chaperone molecules to establish an "open" chromatin environment or a nucleosome-free region (NFR) to facilitate the accessibility for other factors, then initiate subsequent regulatory events. However, the pioneer capacity of GATA2 is inconclusive. Several studies illustrated that GATA2 is simply a non-pioneer TF[30] but others demonstrated it is indeed a pioneer TF[19]. To fully understand the detailed pioneer capacities of GATA2 in regulating nucleosome organization in androgen stimulated prostate cancer cells, it is critical to use integrative approaches combining high-resolution genomic techniques (ChIP-exo and MNase-ChIP-seq) and computational analyses to examine the relationship among GATA2 and nucleosome organization.

In this study, we conducted high-resolution ChIP-exo, MNase-seq, and MNase-ChIP-seq in LNCaP cell model under Vehicle (Veh) and 5α-dihydroxytestosterone (DHT)-treated conditions and generated over three billions multi-omics sequencing data. Here, we exploit the landscape of dynamic nucleosome footprints and quantitatively define functional nucleosome states based on histone marks, genomic regions, nucleosome positioning, spacing, and regularity. We further identify GATA2-associated dynamic nucleosome state switching upon DHT treatment. Using various in vitro assays, we demonstrate that GATA2 initially functions as a PF binding at the edge of a nucleosome in an inaccessible crowding array. Under the DHT-treated condition, GATA2 reconfigures inaccessible to accessible nucleosome state and subsequently, it acts as a master transcription factor either directly or to recruit signaling-specific TFs to enhance oncogenic

Wnt/β-catenin signaling in an AR-independent manner. Our work thus elicits a noncanonical GATA2 pioneer model, providing a structural and mechanistic insight into the dynamics of pioneer factors governed by nucleosome reorganization.

## Results

**Genome-wide identification of nucleosome positioning and spacing.** We conducted multi-omics sequencing profiling in LNCaP cells in Veh and DHT-treated conditions, including MNase-seq for identifying genome-wide nucleosome positioning, spacing, and regularity, MNase-ChIP-seq of H3K4me1, H3K4me2, H3K4me3, H3K27ac, H3K27me3, H3K36me3, and H3K79me2 for detecting enriched histone marks at a nucleosome level, as well as ChIP-ePENS of FOXA1 and GATA2 for detecting 1 bp binding resolution, each with biological replicates (Fig. 1a). In total, we generated over three billons of multi-omics sequencing data to investigate the landscape of nucleosome organization and dynamic reorganization (Supplementary Table 1). A Pearson correlation coefficient between two MNase-seq replicates in LNCaP cells was very high with an $r$ value of 0.95 (Fig. 1b), demonstrating the high reproducibility and good quality of the data. We applied a nucleosome positioning tool iNPS[31] on MNase-seq data, and identified ~12.6 million nucleosomes, in equivalent to 48.6% of the whole genome DNA wrapping on the nucleosomes, which were proportionally distributed in each of 23 chromosomes (Supplementary Table 2). We then plotted a down-sampling saturation curve and showed the data sequencing depth was sufficient to capture whole genome-wide nucleosomes (Supplementary Fig. 1). We further used a nucleosome density map to illustrate the robust correlations of detected nucleosomes in each of 23 chromosomes between two replicates (Fig. 1c), with all $r$ values larger than 0.93. The accumulation of nucleosome dyads with the enrichment of Mono-nucleosome, Di-nucleosomes, Tri-nucleosomes, and Penta-nucleosomes showed relatively high robustness of nucleosome detection with adjacent three nucleosomes in a genome-wide scale. (Fig. 1d). We also observed a range of 150-350 bp with a peak of 187 bp nucleosome spacing, i.e., the distance between two neighboring nucleosome dyads (Fig. 1e).

**Quantitatively defining functional nucleosome states.** To understand the relationship between the nucleosome positioning, spacing, and regularity with eight histone marks which characterize genomic regulatory elements as a promoter, enhancer, repressor, and others, we examined the enrichment of histone marks on positioned nucleosomes in three genomic regions, the Promoter (−1 Kb to 1 Kb away from TSS), the Proximal (−5 Kb to −1 Kb away from TSS), and the Distal (−50 Kb to −5 Kb away from TSS). We observed a distinct distribution of nucleosomes enriched with different histone marks in three genomic regions (Fig. 2a). For example, H3K4me1-nucleosomes were mostly located in far the Proximal and Distal regions and H3K27me3-nucleosomes were in the far Distal region. When looking into MNase-seq read signal distribution of histone marks in different genomic regions, we found the characteristic spacing between nucleosomes. For example, we found a clear spacing between nucleosomes of H3K4me1 and H3K27ac in all three regions, of H3K4me3 in the Promoter/Proximal region, of H3K27me3 in the Proximal/Distal, of H3K4me2 in the Promoter, and of H3K36me3 in the Distal region respectively (Fig. 2b). Further, we observed histone marks and genomic region-specific nucleosome spacing patterns (Fig. 2c). For instance, H3K4me3, H3K27ac, and H3K36me3 have the shortest nucleosome spacing ranging from 170–180 bp in the Promoter region, H3K27me3 has two peaks at 188 bp and 215 bp in the Proximal region, and H3K9me3 and

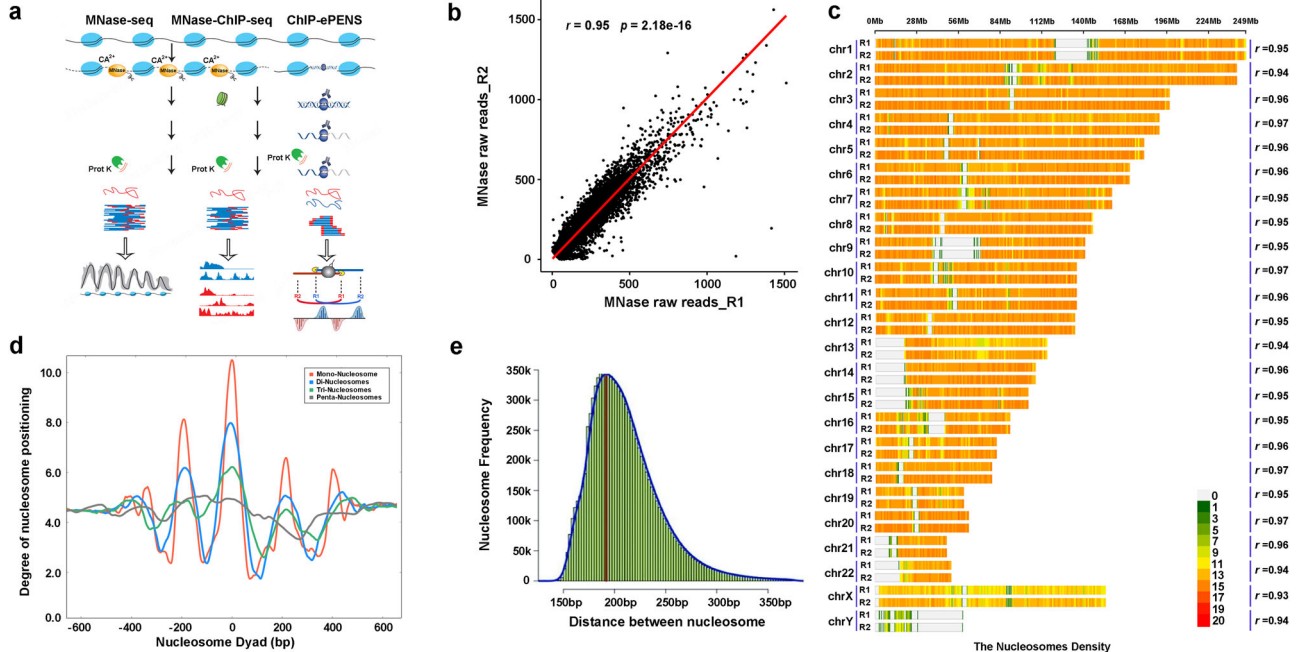

**Fig. 1 Identification of nucleosome positioning and spacing with MNase-seq. a** A scheme for generating multi-omics sequencing data including MNase-seq, MNase-ChIP-seq, and ChIP-exo for detecting nucleosome footprints. **b** A Pearson correlation of raw read counts within a bin size of 200 bp between two MNase-seq biological replicates in LNCaP cells showing a high coefficient value at 0.95 and two-tailed *p* value of 2.18e-16, illustrated by a representative chromosome 10. **c** Nucleosome density distribution between two replicates within a range of 5 Kb in each of 23 chromosomes. **d** Accumulation of nucleosome dyad according to the Mono, Di-, Tri-, and Penta-nucleosomes. Each nucleosome dyad was set as 0 bp and MNasse-seq reads in 600 bp upstream and downstream of each nucleosome dyad were used for plotting the accumulation. **e** The frequency distribution of the distance between adjacent nucleosomes dyad under 400 bp and an overall nucleosome spacing with a peak of 187 bp.

H3K27me3 have a wide peak in the Distal region. Furthermore, we compared the density of histone mark-enriched nucleosomes and gene expression, i.e., No. of nucleosomes per 1000 bp, in three regions, and found active marks, H3K4me1/2/3 and H3K27ac had a higher density than repressive marks, H3K9me3 and H3K27me3 in all three regions (Fig. 2d and Supplementary Fig. 2). Finally, we developed an empirical formula to quantitatively define the nucleosome states (Methods), which takes into account nucleosome positioning, spacing, and histone marks, including the similarity score γ between nucleosome αn +1 and α1 to αn, where γ is calculated by nucleosome position factor λ, the peak area of nucleosome S, the width of nucleosome position W and spacing between nucleosomes d and histone mark factor β. After iteratively running the formula to optimize the parameters, we were able to determine the trajectory of optimized parameters used to define the number of nucleosome states.

Indeed, when further incorporating various genomic features with the trajectory, including histone marks, the number of the grouped nucleosomes, genomic location, degree of positioning, regularity score, and average spacing, we were able to define nine functional states, S1–S9 (Fig. 2e, f and Supplementary Fig. 3). S1 was defined as transcriptional initial due to its location around the up-promoter or 5′TSS region with active H3Kme3/K27ac marks. S2 was defined as an accessible edge due to its mainly located in proximal/distal regions and having 1–4 nucleosomes, a shorter spacing (180.43 bp shortest except S1), and a relatively low regularity score of 9.85. S3 was defined as Alternative primed due to its locating in promoter/proximal regions and having primed or poised marks H3K27me3/H3K4me1/H3K4me2. S4 was defined as Crowding array because of its higher number of 5–20 positioned nucleosomes in an array with the highest average spacing of 210.64 bp and 75.4% of states enriched with H3K9me3 and/or H3K27me3 marks. S5 was identified as well-organized due

to its location in the down-promoter region with the highest regularity score of 21.00. S6 was defined as restricted accessible since a majority (54.1%) of S6 were in the lower average spacing and in proximal/distal regions with H3K4me2/H3K27me3 marks. S7 was defined as a Steady structure due to 69.8% of states were in a distal region with various marks. S8 was defined as Fuzzy due to its lower degree of positioning and the corresponding low regularity score and S9 was defined as Unknown due to its unclear features. S1 and S5 have been extensively studied in previous work[32–34], therefore, we focused on thoroughly examining the functionality of S2, S3, S4, S6, and S7 and their relationship with FOXA1, GATA2, and other TFs and coregulators in the downstream analyses.

**Dynamic nucleosome states switching.** We extended our quantitative modeling of nucleosome states on MNase-seq and MNase-ChIP-seq data in LNCaP cells under DHT-treatment conditions. Intriguingly, we obtained the same functional nucleosome states, demonstrating the validity and broadness of our quantitative definition. When comparing the changes in nucleosome states before and after DHT treatment, we found the numbers of S2, S3, and S6 increase while the numbers of S4 and S7 decrease (Fig. 3a). Interestingly, we found dynamic switches among different nucleosome states upon DHT treatment. Sankey's diagram showed over 72.1% of S4 have been switched to other states including 45.4% of them turning into S3, while 42.3% of S3 have been changed to S2, and the number of S2 increases to 202.4% in the DHT-treatment condition (Fig. 3b). Next, we wanted to investigate what transcription factors (TFs) and coregulators could potentially instruct these dynamic nucleosome state switches particularly from two condensed states, S4 (Inaccessible) or S3 (Alternative primed) to others. We downloaded

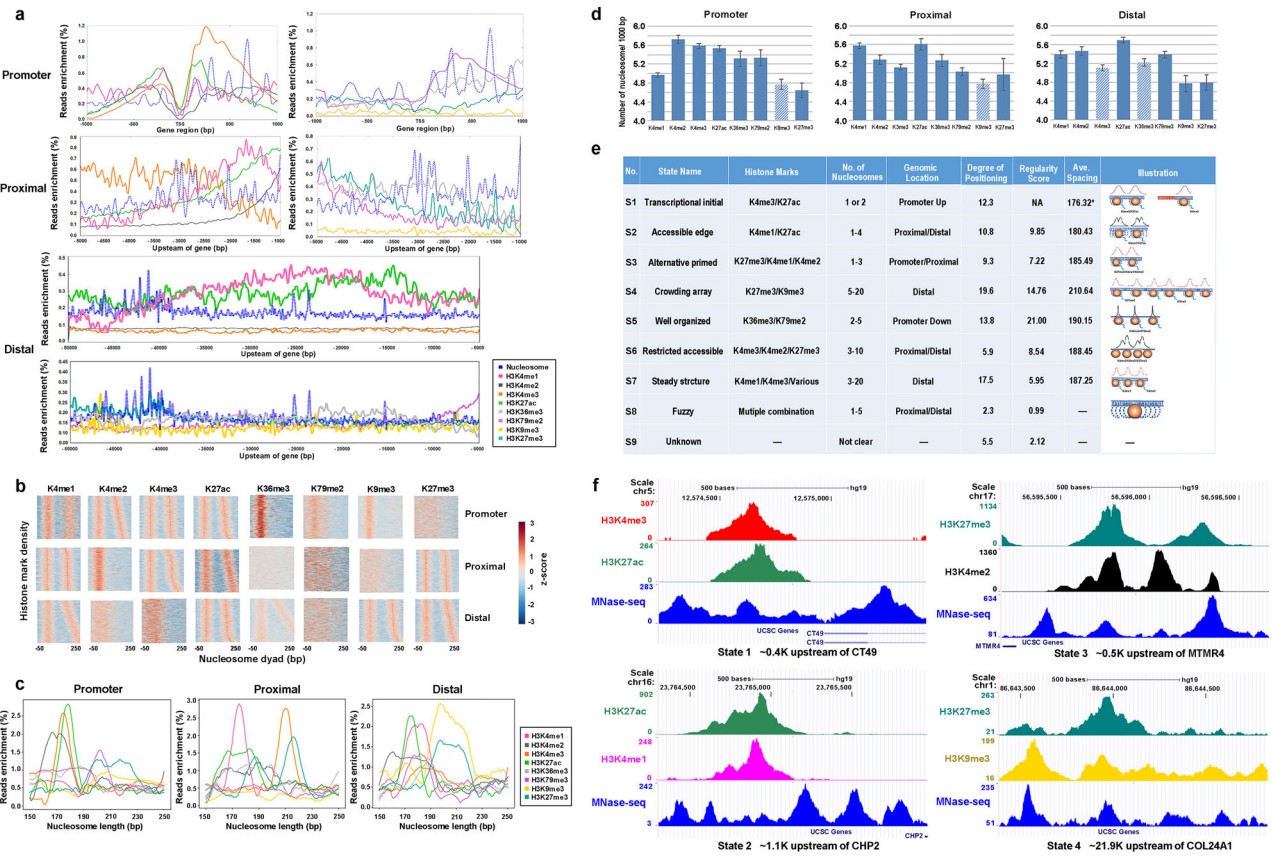

**Fig. 2 Quantitatively defining functional nucleosome states. a** A distribution of Nucleosome (blue), H3K4me1 (pink), H3K4me2 (gray), H3K4me3 (orange), H3K27ac (green), H3K36me3 (light gray), H3K79me2 (purple), H3K9me3 (yellow), H3K27me3 (cyan) in three genomic regions, Promoter (−1 Kb ~1 Kb of transcriptional start site), Proximal (−5 Kb ~1 Kb), Distal (−50 Kb ~ −5 Kb) and each line showed an average for biological replicates. **b** MNase-seq read signal distribution heatmap of histone marks in different genomic regions showing the characteristic spacing and regularity between nucleosomes. **c** A plot of enrichment curve showing histone mark and genomic region specifically nucleosome spacing pattern. **d** The density of histone mark-enriched nucleosomes, i.e., No. of nucleosomes per 1000 bp, in three regions and the shadowed bars mean a lower detection of a specific mark in this region. Each bar represents the mean value with the standard deviation as error bars. **e** Quantitative definition of functional nucleosome states. **f** The visualization for functional nucleosome states, S1–4.

many publicly available ChIP-seq data of PFs, TFs, and coregulators and examined their differential binding patterns[35–38] (Fig. 3c and Supplementary Fig. 4). As expected, we found FOXA1 showed a significantly lost binding from S4 to S3 or S2 accompanied by a lost linker H1. This finding is consistent with many other studies[39,40], where FOXA1 competes with its canonical binding motif with H1 to enhance nucleosome accessibility (Supplementary Fig. 5). We also found the bindings of GATA2, HOXB13, RUNX1, and TLE3 were lost, implicating their potential pioneer capacities of opening condensed nucleosomes. We then particularly examined the expression level of genes associated with nucleosome states switching from S4 to relatively accessible states (RAS) including S2, S3, S7, and nucleosome-free region (NFR) named as RAS1; and from S3 to relatively accessible states including S2, S6, and NFR named as RAS2. Interestingly, we found a majority (>80%) of the genes associated with differential binding of FOXA1, GATA2, and H1 were upregulated, indicating both FOXA1 and GATA2 play pioneer roles capable of dynamically reprogramming nucleosome accessibility, resulting in gene activation (Fig. 3d and Supplementary Figs. 6, 7). Furthermore, we found there were 257 and 293 unique GATA2 genes with only GATA2 binding but no other analyzed TF bindings from S4 to RAS1 and S3 to RAS2 respectively, suggesting that this subset of GATA2 genes can independently exert a pioneer function upon androgen stimulation (Fig. 3e).

**GATA2-associated dynamic nucleosome states switching**. To elucidate the pioneer capacity of GATA2, we conducted ChIP-ePENS of GATA2 in both Vehicle (Veh) and DHT-treated LNCaP cells and used ePEST to identify GATA2 binding borders at one-base resolution[21] (Supplementary Fig. 8 and Supplementary Table 3). We identified a total of 32,342 and 27,613 border composed sites (BCSs) as binding footprint boundaries in Veh and DHT-treatment respectively. Interestingly, we found paired border sites (PBSs) of GATA2 borders followed a bimodal distribution with a 13–14 bp gap, similar to the distribution of FOXA1 borders with an 11 bp in our previous study[41] (Fig. 4a, Supplementary Fig. 9, and Supplementary Data 1). About 42.1% of GATA2 borders were associated with the nucleosome states in Veh and dropped to 36.2% in DHT-treated cells (Fig. 4b, Supplementary Figs. 10, 11, and Supplementary Data 2, 3). GATA2 borders on S2 showed a dramatic decrease at 45.5% upon the DHT treatment (Fig. 4c). By plotting the accumulating distribution of borders around the nucleosome dyad, we found more than 80% of GATA2 borders were located ~50–60 bp on the edge of the nucleosome for S4 and S3, and the peaks tended to spread wider in the DHT treatment, while a majority of GATA2 borders were located in the middle of nucleosomes for S6 and S7 (Fig. 4d and Supplementary Fig. 12). Two examples demonstrated GATA2-associated nucleosomes switching from inaccessible to accessible states while maintaining its binding (Fig. 4e and Supplementary Fig. 13).

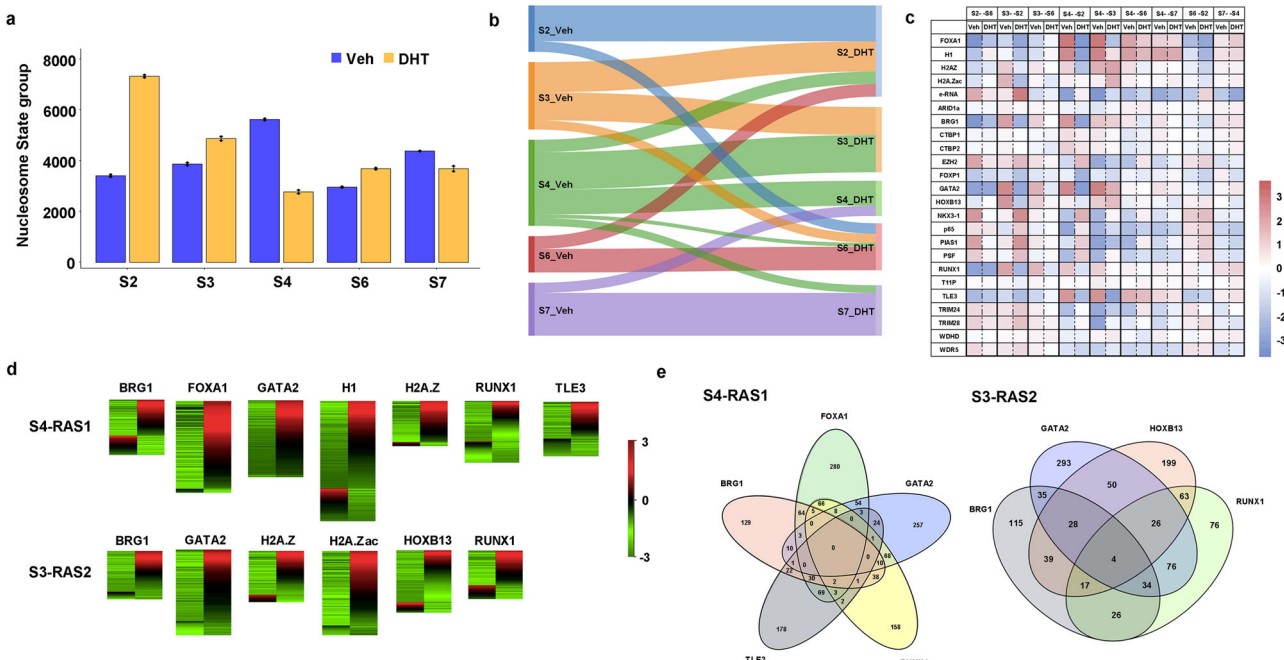

**Fig. 3 Dynamic nucleosome states switching from Veh to DHT-treated conditions. a** A histogram showing the number of nucleosomes in S2, S3, S4, S6, and S7 in Veh and DHT-treated LNCaP cells, respectively. **b** Nucleosome states switch in the same genomic region comparing Veh and DHT-treated conditions. **c** Differential binding of FOXA1, GATA2, H1, other TFs, and coregulators in a specific nucleosome state in Veh and DHT-treated conditions. **d** Differential gene expression for those genes associated with S4-RAS1 (relatively accessible states 1) and S3-RAS2 (relatively accessible states 2) switching accompanying differential binding of the enriched factors. **e** Overlapping genes of the enriched TFs associated with switched states of S3 and S4 to RASs.

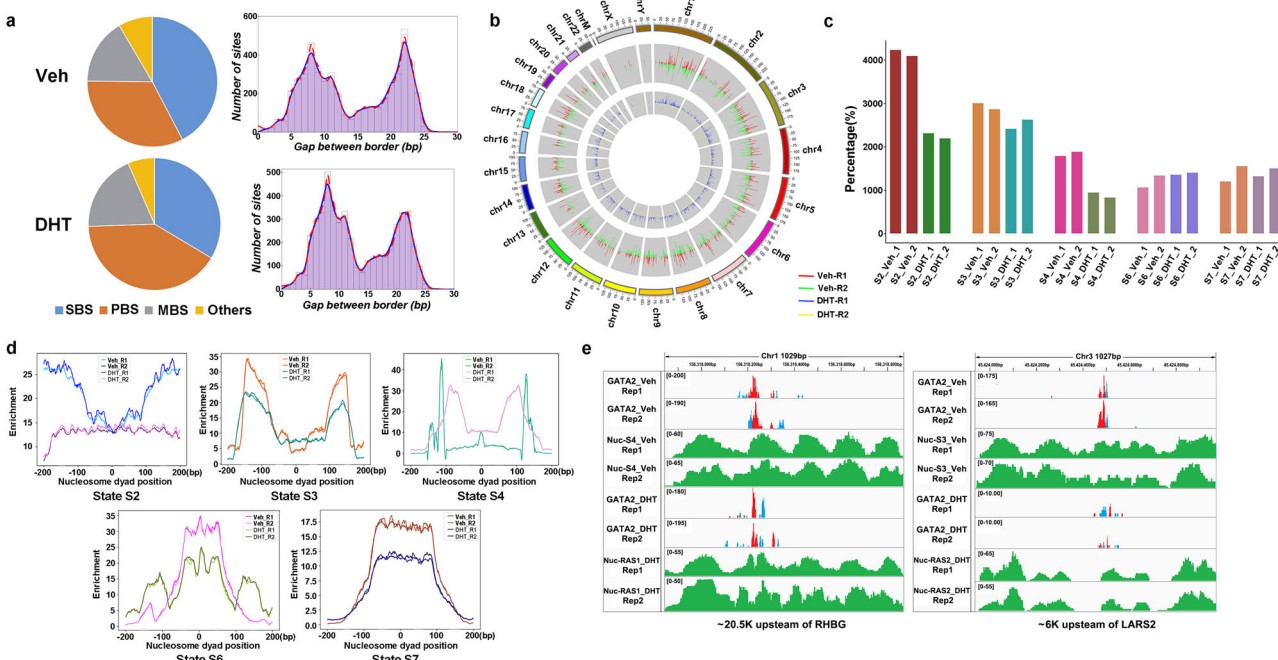

**Fig. 4 GATA2-associated functional nucleosome states in Veh and DHT-treated conditions. a** Identification of GATA2 borders and a distribution of gaps between borders showing a bimodal pattern for each of two biological replicates. **b** GATA2 border distribution within different chromosomes in the conditions of Veh and DHT treatment. **c** The number of GATA2 borders located on nucleosomes and distributed on different nucleosome states for each of two replicates. **d** An enrichment plot of GATA2 borders in different nucleosome states for each of two replicates. **e** Two screenshots showing the changes of GATA2 borders along with nucleosome states switching.

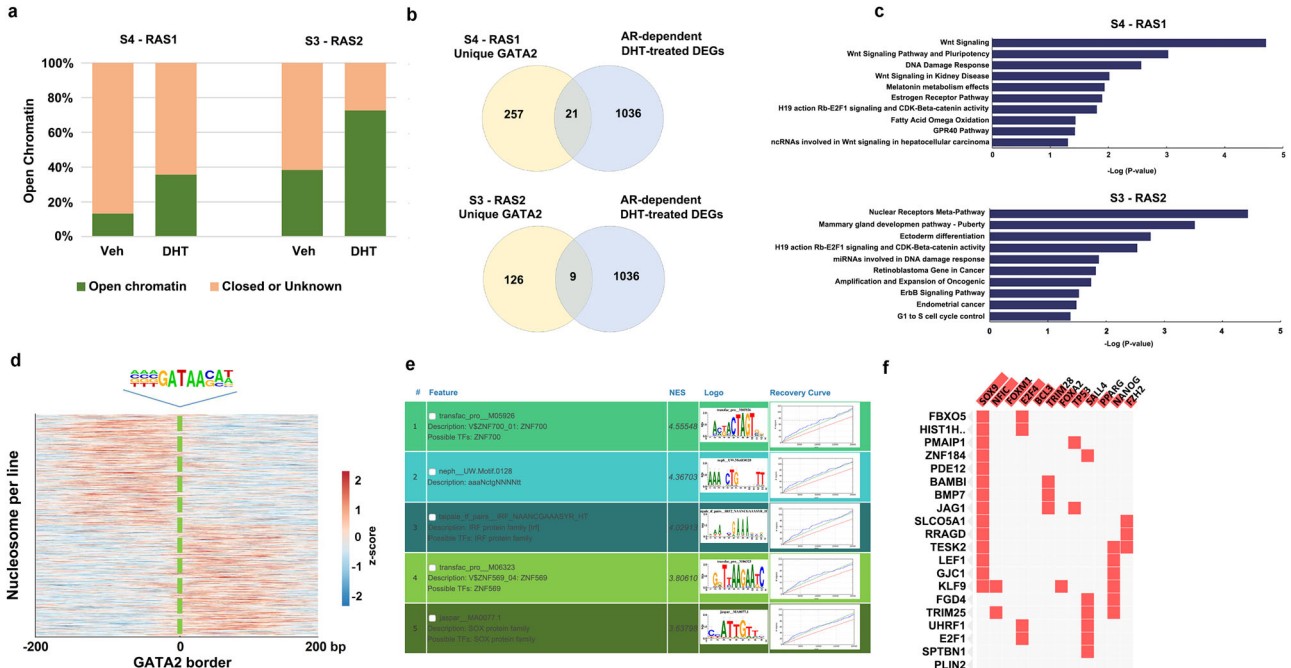

**Fig. 5 GATA2 governing nucleosome states switching from Veh to DHT-treated conditions. a** Open chromatin was detected by ATAC-seq for GATA2 binding sites associated with nucleosome states switching in Veh and DHT-treated conditions respectively. **b** Overlapping between GATA2-associated nucleosome state switching genes and AR-regulated differentially expressed genes in DHT-treated condition. Differentially expressed genes were detected with $p$ value <0.05 and | log2(Fold Change)| > 1. **c** KEGG pathway analyses showing the top pathways in S4-RAS1 and S3-RAS2 respectively. **d** The location of the GATA2 motif between ChIP-exo borders relative to the density map of MNase-seq reads for positioned nucleosomes. **e** The prediction of regulatory features and modules by i-cisTarget illustrating several top enriched TF motifs in unique GATA2-associated nucleosome state switching genes. **f** SOX9 is the most enriched co-binding TF on WNT signaling genes identified by a publicly available database that collected all ChIP-seq of TFs from ENCODE and ChEA.

We further examined the open chromatin changes for both GATA2-associated nucleosome states switching by ATAC-seq data and observed a great increase from 11.2 to 37.1% for S4-RAS1 and 38.6 to 72.3% for S3-RAS2, respectively (Fig. 5a and Supplementary Fig. 14). Further, only 21 (7.6%) of 278 genes associated with unique GATA2-governed S4-RAS1 switching and only 9 (6.7%) of 135 genes associated with unique GATA2-governed S3-RAS2 switching were overlapped with 1036 AR-dependent DHT-treated differentially expressed genes, respectively, suggesting that the vast majority of GATA2-governed dynamic nucleosome states switching are independent of AR signaling (Fig. 5b and Supplementary Figs. 15, 16). KEGG pathway analysis further identified WNT signaling and nuclear receptor meta pathways were the top pathways for S4-RAS1 and for S3-RAS2 respectively (Fig. 5c). GATA2 strongly preferred binding at the edge of nucleosomes and were almost equally distributed on both sides (Fig. 5d). Remarkably, ZNF700, IRF protein family, ZNF569, and SOX protein family were found as the top enriched motifs in GATA2-governed S4-RAS1 switching (Fig. 5e) and SOX9 was also identified as a potential co-binding TF on WNT signaling genes by a publicly available database collected all ChIP-seq of TFs from ENCODE and ChEA[42] (Fig. 5f and Supplementary Fig. 17). Taken together, our data suggested a noncanonical pioneer model of GATA2 that it initially functions as a PF binding at the edge of a nucleosome in an inaccessible crowding array; under the DHT-treated condition, it reconfigures inaccessible to accessible nucleosome state; subsequently, it acts as a master transcription factor either directly or to recruit other signaling-specific TFs to enhance WNT signaling in an AR-independent manner.

**GATA2 in mediating nucleosome dynamics and enhancing WNT signaling pathway**. To substantiate this model, we conducted various in vitro assays on 20 Wnt/β-catenin signaling genes selected from GATA2-governed S4-RAS1 switching. Competitive nucleosome-binding assays including in vitro nucleosome-binding and electrophoretic mobility shift assays were designed to detect a binding range of nucleosome position with GATA2 binding border at 0, 41, 65, and 85 bp. The 65 bp DNA showed the highest supershift of the others, confirming that GATA2 prefers binding at the edge of nucleosomes (Fig. 6a and Supplementary Fig. 18). Open chromatin assays further demonstrated that 17 of 20 Wnt/β-catenin signaling genes showed an increase in chromatin opening upon DHT treatment (Supplementary Fig. 19). Our genome-wide ATAC-seq data showed that a majority of 413 GATA2-governed S4/3-RAS1/2 switching genes significantly reduced the chromatin accessibility and overall 29.6% of open chromatin regions on a genome-wide scale were lost after knockdown GATA2 gene (Fig. 6b and Supplementary Figs. 20, 21). Together, we provided several lines of evidence to support the notion that GATA2 was involved in regulating chromatin accessibility and nucleosome reorganization. Furthermore, we used siRNA to knock down GATA2 in LNCaP cells to create a siGATA2 subline and measured the gene expression changes by RT-qPCR. We found that 14 of 20 genes in siGATA2 vs siCtrl LNCaP cells were downregulated under the DHT-treated condition (Fig. 6c), suggesting that GATA2 regulates Wnt/β-catenin signaling gene expression levels. Collectively, our results revealed a dual pioneer and master role of GATA2 in mediating nucleosome dynamics and enhancing downstream Wnt/β-catenin signaling in an AR-independent manner (Fig. 6d).

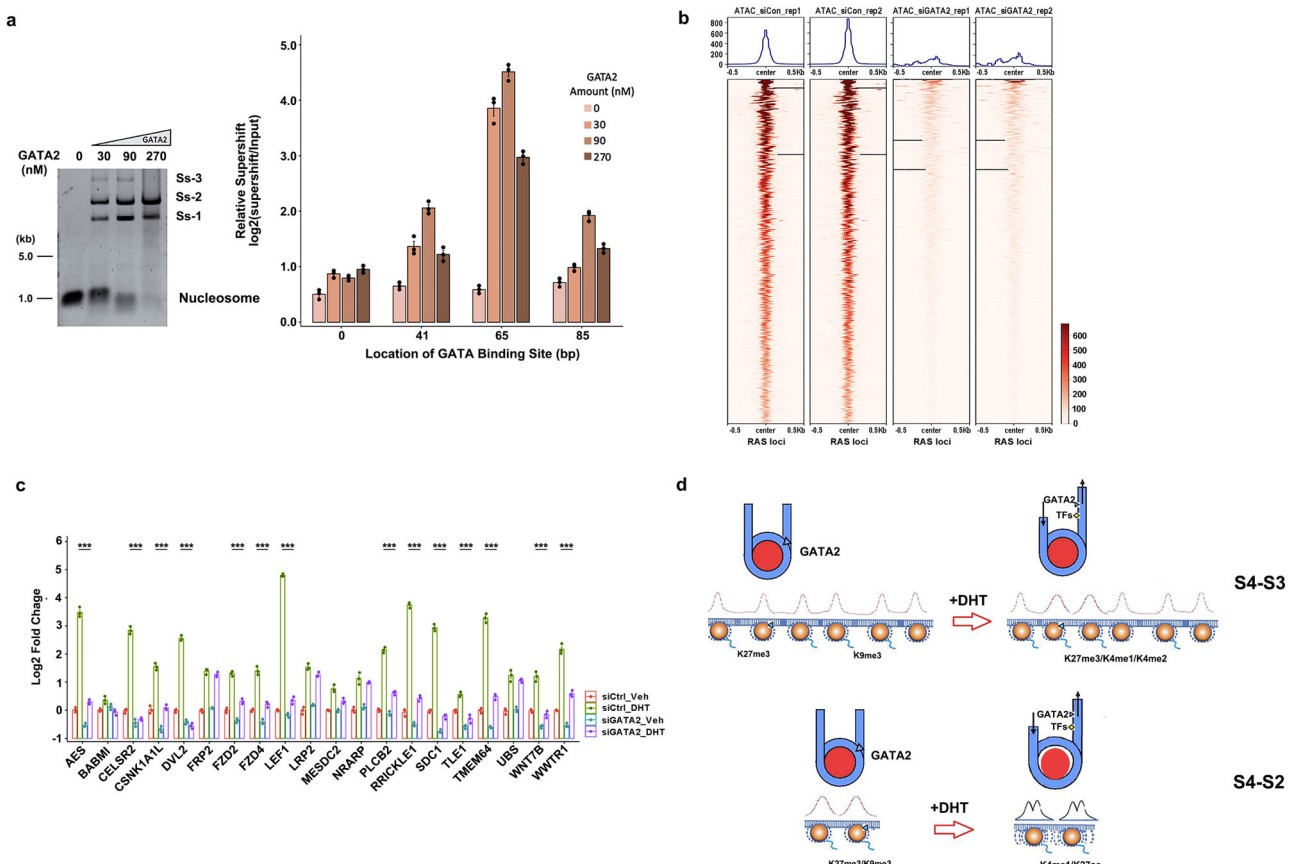

**Fig. 6 Performing functional examinations on GATA2-mediated nucleosome states switching and GATA2-regulated Wnt/β-catenin signaling genes.** **a** Nucleosomes containing 16 different sequences were bound to increase amounts of GATA2 protein and separated by native PAGE. Nucleosomes with 30, 90, and 270 nM of GATA2 and the major supershift bands (Ss-1, Ss-2, Ss-3) were indicated and quantified by qPCR relative to the input nucleosomes. Experiments were repeated independently three times with similar results. Each bar represents the mean value with the standard deviation as error bars. **b** ATAC-seq reads density maps around GATA2-associated RAS loci within 0.5 Kb up/down-stream in siCtrl LNCaP cell line and siGATA2 subline. **c** RT-qPCR measurement on a set of 20 Wnt/β-catenin signaling genes in siGATA2 before and after DHT treatment. The center for the error bars represents the mean value and the error bars represent the standard deviation with three experiments with a "***" showing $p < 0.001$ (two-tailed Student's $t$-test) between Veh and DHT groups for each gene loci. **d** A working model showing GATA2 initially binds at the edge of a nucleosome at a crowding array (S4) and reconfigures it to an accessible primed state (S3/S2) upon androgen (DHT) stimulation, subsequently directly regulates or recruits other TFs to coregulate Wnt/β-catenin signaling genes. Source data are provided as a Source Data file.

## Discussion

In this study, we systematically defined genome-wide functional nucleosome states with a quantitative method (Fig. 2e). One major advantage of our method is to fully utilize the high-resolution nucleosome level genomic data including MNase-seq and MNase-ChIP-seq. Intriguingly, we were able to obtain the same functional nucleosome states in DHT-treated LNCaP cells as in untreated cells when applying the method to the data, demonstrating the validity and broadness of our quantitative definition. Although numerous studies have revealed the basic principles of nucleosome organization and its dynamics[3,4], our work clearly filled a knowledge gap in the field since most of the previous work were focused on qualitatively defining the nucleosome states without systematically providing the trajectory of clear quantitative cutoff thresholds[38], or on examining nucleosome landscape in specific genomic regions[4].

More importantly, these functional nucleosome states could be further used to elicit the pioneer capacity of any TFs including known PFs by integrating with one-base resolution ChIP-ePENS or ChIP-exo data. In theory, our integrative approach can accurately define the pioneer capacity of any known PFs or distinguish the pioneer factors from non-pioneer factors by comparing two or more different biological conditions. This statement is attested

by the following four foundations of our approach: (1) we identify the PF/TF-associated condensed nucleosome states; (2) we identify the PF/TF binding borders within the nucleosomes; (3) we determine whether the PF/TF is accompanying the nucleosome switches under at least two biological conditions; and (4) we perform competitive nucleosome-binding assays to validate the pioneering capacity. Although our approach is able to define the pioneer functionality of any TFs, we are cautious that the PF functionality and capacity should be interpreted tightly with the specific biological context. Nevertheless, our approach highlights the importance of utilizing the high resolution of high throughput genomic data in elucidating the pioneer function of TFs.

Remarkably, we found almost half (42.1%) of GATA2 was bound on nucleosomes (combined all states) in untreated LNCaP cells and 51.8% of these GATA2-associated nucleosomes were switched to more accessible nucleosomes or free nucleosome regions in DHT-treated cells (Fig. 4b), suggesting that this subset of GATA2 might function as a pioneer factor in the hormone-induced context. We unexpectedly found this GATA2 pioneer action exerts in an AR-independent manner and regulates specific downstream signaling pathways (Fig. 5b, c). It seems that GATA2 further acts as a master transcription factor either directly or to recruit other signaling-specific TFs to the chromatin to regulate

the Wnt/β-catenin pathway upon androgen stimulation (Fig. 6). This data is in stark contrast with the archetypical pioneer function of FOXA1 such that FOXA1 opens the condensed chromatin to mainly serve for an AR binding activity under DHT-treated conditions (Supplementary Figs. 22, 23). Collectively, our data support a noncanonical pioneer GATA2 model and elicit the pioneer capacity of GATA2 action in hormone-induced prostate cancer cells.

Despite that previous studies[19,25–27] have demonstrated the pioneer functionality of GATA2 in hormone-induced prostate cancer cells, all of these studies emphasized on the pioneering role of GATA2 in activating or enhancing AR-dependent gene transcription. By contrast, our results illustrated a pioneer function for GATA2 regulation in which it regulates oncogenic Wnt/β-catenin signaling by circumventing AR signaling. Our finding that GATA2 exerts an AR-independent functionality in promoting aggressive prostate cancer is consistent with a previous study that GATA2 regulates a core subset of clinically relevant genes in an AR-independent manner[43].

In summary, we provided a quantitative model of defining functional nucleosome states to the community. We also conducted a detailed examination of the pioneer capacity of GATA2 in regulating dynamical nucleosome reorganization in hormone-induced prostate cancer cells, and further implicated GATA2-mediated Wnt/β-catenin signaling in conferring aggressiveness in prostate cancer. Our work may provide a rationale for targeting GATA2 downstream signaling as a therapeutic strategy to treat advanced prostate cancer. Our work also offers a structural and mechanistic insight into the dynamics of pioneer factors governed by nucleosome reorganization.

## Methods

**MNase-ChIP-seq and MNase-seq**. MNase-ChIP-seq and MNase-seq protocols were performed according to previous studies[44]. In brief, LNCaP cells were exposed to 10 nM DHT or DMSO (Veh) for 4 h. Mono-nucleosomes with solubilized chromatin was achieved by MNase digestion of 2 min at 37 °C, then immuno-precipitated with antibody-conjugated magnetic beads. DNA is phenol extracted and ethanol precipitated. Libraries were prepared from isolated DNA and sent for sequencing on the Illumina HiSeq3000 at the UTHSA sequencing core. All samples were performed in biological replicates. Antibodies include: H3K4me1 (ab8895) 1:500 dilution, H3K4me2 (ab7766) 1:250 dilution, H3K27ac (ab4729) 1:500 dilution, H3K27me3 (ab6002) 1:250 dilution, H3K36me3 (ab9050) 1:250 dilution, H3K79me2 (ab8898) 1:250 dilution from Abcam (Cambridge, MA). H3K4me3 (07-473) 1:500 dilution, H3K9me3 (17-10242) 1:250 dilution from Millipore (Upstate).

**ChIP-ePENS**. A modified ChIP-exo protocol for TFs was performed as following steps[21]: Cells were fixed with 1% formaldehyde for 10 min at room temperature and chromatin was sonicated and incubated overnight with 2–4 μg antibodies against GATA2(sc-9008, Santa Cruz) 1:250 dilution with biological replicates. T4 DNA polymerase, T4 PNK, and Klenow DNA Polymerase were used together for end polishing. The ligation step was performed with 1 mM dithiothreitol. Protein A Dynal magnetic beads were washed using modified RIPA buffer (50 mM Tris-HCl pH 8.0, 1 mM EDTA, 0.25% sodium deoxycholate, 1% NP-40, 0.5 M LiCl) followed by Tris pH 8.0 twice during each step. The library was amplified with only 10–12 cycles and prepared without gel-based size selection. Paired-end sequencing (50 bp) was performed by Illumina HiSeq2500.

**ATAC-seq**. About 50,000 cells were resuspended in cold ATAC-seq resuspension buffer (10 mM Tris-HCl pH 7.4, 10 mM NaCl, and 3 mM MgCl2). Cell nuclei were then prepared by incubation in 50 μl of ATAC-seq resuspension buffer containing 0.1% NP-40, 0.1% Tween-20, and 0.01% digitonin on ice for 3 min. After centrifugation, nuclei were resuspended in 50 μl of transposition mix (25 μl 2× TD buffer, 2.5 μl Nextera Tn5 transposase (Illuminar), 16.5 μl PBS, 0.5 μl 1% digitonin, 0.5 μl 10% Tween-20, and 5 μl water), and incubated at 37 ℃ for 30 min in a thermomixer with shaking at 1000 rpm. Transposed fragments were then purified with a Zymo DNA Clean and Concentrator-5 Kit. All libraries showed sufficient amplification after the five pre-amplification cycles and were quantified using the KAPA Library Quantification Kit. Libraries were then sequenced using Illumina Novaseq 6000 at the Duke sequencing core. All samples were performed in biological replicates.

**Data mapping**. MNase-seq, MNase-ChIP-seq, and ChIP-ePENS sequencing datasets were generated in LNCaP cells with both Veh and DHT-treated conditions. Raw sequence reads were aligned against the human genomic sequence (hg19) using bowtie2 (version 2.2.8) with -v 3 -k 2 -m 1 -I 20 -X 400 for ChIP-seq data. Only uniquely mapped reads were used for further downstream analysis.

**Detection of nucleosome positioning**. We applied iNPS[31], which uses a Laplacian of Gaussian convolution model to obtain smooth estimates of the stringency of nucleosome positioning, on MNase-seq data to detect nucleosome centers (dyads) and robustly estimate the degree of positioning. The kernel bandwidth w is a key parameter to control the smoothness of the stringency profile. We initially chose w = 30 as suggested in ref. [39] to conduct the calculation, then adjust it to make sure that it provides sufficient smoothing for the particular data without sacrificing the sharpness of the positioning estimate. We assigned nucleosomes to 21,319genes with their longest isoforms in which each gene was divided into three genomic regions: Promoter region of −1 Kb to 1 Kb around the transcriptional start site (TSS), Proximal region of −5 Kb to −1 Kb upstream of TSS and Distal region of −50 Kb to −5 Kb upstream of TSS. We used MACS2[44] to identify enriched peaks of various histone marks using nonmodel and shift options in order to remove a technical bias and peak shifting for ChIP-seq data.

**Nucleosomesgrouping and states classification**. The degree of positioning describes how well the nucleosome is positioned in the cells' population and the regularity score indicates the periodical feature of a nucleosome array, which is measured by calculating power spectral density with an interpolating method and Welch's method for the nucleosome states' array.

For quantitatively defining the nucleosome states, we utilized the following features related to nucleosomes, nucleosome positioning and spacing, histone marks, and the similarity ratio between continuous nucleosomes.

$$\lambda_i = S_i/W_i + \omega(\mu - d_i) \tag{1}$$

$$\beta_i = \sum_{k=1}^{n} \frac{dk}{dx} a^k b_i \tag{2}$$

where $\lambda_i$ represents the nucleosome positioning and spacing, as $S_i$ is the peak area of nucleosome $N_i$, $W_i$ is the width of nucleosome position, $\omega$ is the weight for the spacing factor, $\mu$ is the average spacing in a specific area and $d_i$ is the actual spacing between nucleosomes. $\beta_i$ represents the histone mark state factor, as $\frac{dk}{dx}$ is the weight of a specific histone mark k, while $a^k$ is the relative number of reads of histone mark k and $b_i$ is the relative number of reads of nucleosome $N_i$

$$\gamma = (\lambda_i\beta_i - \lambda_{i+1}\beta_{i+1})/\sum_{n=1}^{i}\lambda_i\beta_i \tag{3}$$

where $\gamma$ calculates the similarity ratio between nucleosome $N_{i+1}$ and $N_1$ to $N_i$. If $\gamma$ among calculated nucleosomes is lower than 10%, the nucleosomes were merged into the same group.

We defined nucleosome dyad position, degree of positioning, regularity score, and histone peak signals as grouped nucleosome profiles. We then performed K-means clustering on grouped nucleosome profiles to obtain distinct classes of nucleosome states.

**Identification of GATA2 borders**. Border-calling of ChIP-ePENS data was conducted by ePEST (version 1.0) with the parameter of -D True -p 1e-8 -R 25 -c 0.05 -k 2.0 -o. The ePEST algorithm was specifically designed for ChIP-ePENS[40] and depending on a statistical evaluation of Chernoff inequity on exo-5′-end reads and r-scan statistic method for peak-calling on son-3′-end reads, Border-calling was conducted specifically within these binding regions and borders were finally assigned into each individual binding site by a graph-based strategy. A GATA2-associated gene was defined as the closest gene of GATA2 border bindings and each GATA2 border pair was assigned to only one gene according to the order of the following criterion: gene body region (TSS~ TES of a gene), promoter region (TSS~ −1 Kb upstream of a gene), proximal region (−5 Kb ~ −1 Kb upstream of a gene), then distal region (−50 Kb ~ −5 Kb upstream of a gene) and no associated gene.

**Differentially expressed gene andATAC-seq analysis**. RNA-seq data were aligned by STAR (version 2.5.3) with default parameters. The differentially expressed genes were performed by HTseq-count (version 0.9.1) and DESeq2 (version 1.10.1) with thresholds of $\log_2(|\text{folder change}|) >1$ and p values < 0.05. ATAC-seq data of LNCaP cells were downloaded from the Gene Expression Omnibus (GSE105116). ATAC-seq peaks were called using HOMER (version 4.10) find peaks localSize 50000 -size 150 -minDist 50 –fragLength 0 -style dnase. Differential accessibility was called using DESeq2 and hyper- and hypo-accessible peaks were defined with a $|\log_2 FC| >1$ and an adjusted p value < 0.01.

**siRNA assay**. Silencer® Select siRNAs of GATA2 were obtained from Thermo Fisher Scientific (Catalog #4392420, Santa Clara, CA). Each sample was performed in triplicates with siRNA of different targets. For transfection of siRNA oligos, cells

were seeded in six-cell plates with Lipofectamine® RNAiMAX Transfection Reagent for 48 h. The knockdown efficiency was detected by qPCR.

**In vitro nucleosome-binding and electrophoretic mobility shift assays**. The nucleosome-binding assay was performed as following[22]: In vitro nucleosomes were generated from H2A/H2B dimer and H3.1/H4 tetramer (NEB). Synthesized double-stranded DNA sequences were mixed at equal molar amounts and then added to histones at octamer/DNA molar ratios of 1.5:1 in 2 M NaCl. Nucleosomes were reconstituted through salt gradient dialysis and further purified by 7–20% sucrose gradient centrifuge and concentrated by 50,000 centrifugal filter units (Millipore, Amicon ultra). The protein-nucleosome-binding assays were carried out with the purified nucleosomes mentioned above and human full-length recombinant GATA2 protein (Abcam catalog no. ab134866) in a 7 μL DNA binding buffer and incubated for 30 min. Protein binding was analyzed by non-denaturing polyacrylamide gel electrophoresis with ethidium bromide staining.

**Open chromatin assay**. Open chromatin assay was performed using a Chromatin Accessibility Assay Kit (ab185901) from Abcam (Cambridge, MA)[45]. Briefly, LNCaP cells were lysed and chromatin was extracted by adding lysis buffer to cell pellets and incubating for 10 min. Chromatin pellets were centrifuged and resuspended in a nuclease reaction mix and then added to a stop solution and incubated with Proteinase K. Accessible fragments were then purified by DNA binding columns and gene targets were analyzed with PCR by comparing the nuclease-treated condition and the no-nuclease control.

## Data availability

The data that support this study are available from the corresponding authors upon reasonable request. The MNase-seq, MNase-ChIP-seq, ChP-ePENS of GATA2, and ChIP-seq of H1 data generated in this study have been deposited in the GEO database under accession code GSE148935 and GSE182529. Source data are provided with this paper.

## Code availability

All codes used in this study are available on Github (www.github.com/tianbao365/Nuc-PF) and Zenodo (https://zenodo.org/badge/latestdoi/296872753).

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

## Acknowledgements

We thank UTHSA Next Generation Sequencing Facilities (1S10OD021805) for services rendered for the production of ChIP-ePENS, MNase-seq, MNase-ChIP-seq, and thank Duke sequencing core for services rendered for the production of ATAC-seq and ChIP-seq data. We are grateful to Dr. B Frank Pugh of the Department of Molecular Biology and Genetics at Cornell University for reading the manuscript and providing suggestive comments. This project was partially supported by grants from NIH R01GM114142 (V.X.J.), U54CA217297 (V.X.J. and Q.W.), and R01GM120221 (Q.W. and V.X.J.).

## Author contributions

V.X.J. conceived the project. V.X.J. and Q.W. conceived the functional validations. Q.W. provided the critical inputs to the project. T.L, Q.L., Z.C., and F.H. conducted the experiments. T.L. and Q.L. performed the data analyses. V.X.J., T.L., Q.L., Z.C., and Q.W. wrote the manuscript, with all authors including K.F., X.F., and F.H. contributing to the writing and providing the feedback.

## Competing interests

The authors declare no competing interests.
