## [Peer Review File · Nature Communications]

Dynamic nucleosome landscape elicits a novel noncanonical GATA2 pioneer modelREVIEWER COMMENTS

Reviewer #1 (Remarks to the Author):

Tianbao Li and colleagues present a multi-functional genomics approach to characterize changed chromatin/nucleosomal states upon hormone activation. Based on nucleosome maps and histone modifications they categorize nucleosomal states in a prostate cancer cell line. Upon DHT treatment they observe changes in these states and are able to correlate these with altered transcription factor binding. They single out GATA2 after comparing state changes with expression changes and postulate pioneer factor activity. They then apply ChIPsPENS mapping of GATA2 and observe binding to the nucleosomal borders as well as an accessibility increase after DHT treatment. Interestingly, GATA2 seems to have pioneering activity that is independent of the androgen receptor, a mode of action that might involve SOX9 recruitment. Using EMSA they claim a particular binding mode of GATA2 to nucleosomal DNA and demonstrate on candidate genes altered gene expression after GATA2 knockout.

The manuscript contains a lot of interesting data and I appreciate a lot the novel conceptual approach. However, result presentation is substandard to state it mildly. Accordingly it is very difficult to judge the validity of this interesting approach. This manuscript needs massive overhaul to meet decent reporting standards.

Major issues

1) Reporting/Description of what exactly has been done is missing in Materials and Methods and Figure Legends.

For example, Figure 1b: Pearson correlation coefficient has been calculated on what metric? "# of % Aligned bases" is an unknown magnitude to me. Please explain better. Furthermore, 4 decimal places for r are pointless. Anyway, to claim reproducibility for some data it would be important to know how robust the biologically relevant signal is. In terms of nucleosomes this could be the robustness of dyad positions. Anyway, details about what kind of data has been processed how is missing in almost if not all legends.

Most if not all figures lack information on how many instances and/or replicates have been analysed. e.g. how many promoters in Figure 2a, do we see here an average of replicate experiments? This information is crucial and missing in essentially all figure panels. It would be interesting to see in 4d the tracks for the different replicates.

The description of the state classification method is completely cryptic. At which point and based on which metric and method does classification actually take place? It would be great to have a code repository not only for this method but most of the procedures applied in this study as it appears impossible to understand in detail what the authors did.

Bioinformatic procedures such as mapping, nucleosome calling should be reported in detail with all parameters, version numbers of softwares as well as the versions of both genome and genome annotations used.

2) Figures are incomplete and/or contain unnecessary, unscientific elements

Fig 1c: 3D piechart with a mysterious arrow on the right side. 3D representations of piecharts are unnecessary. The figure contains too much visual information for the simple message "50% of the genome is NDR". However, this information is problematic as this

number appears way too high. I am actually concerned that technical reasons prevent the authors from calling nucleosomes on large parts of the genome.

Fig 1d: why put an arrow (with shadow) and not a horizontal dashed line at 187 bp?

Axes labels are missing (e.g. Figs 2a, 2b 2c), legends are barely readable (e.g. Fig 2c), scales/ranges for colour codes missing (Figs 2c, 2d)

Fig 6d: given the skewed nature of ratios, the enrichments/fold changes in 6d should be displayed on log scales.

What is the nature of error bars in Figs 2d/6a/6d?

What is the ordering principle in the Fig 2b heatmaps? what are the reference points for these heatmaps?

3) Biological replicates

In none of the analyses it is clear how information of biological replicates has been incorporated. In fact, for some analyses e.g. state changes it would be very important to see how consistent the state changes are by having separate analyses for the (two?) replicates.

4) The EMSA is an important experiment but without free DNA it is impossible to judge whether nucleosomes have been bound or not. please include a) a lane just with free DNA and lanes with free DNA and GATA2. Why do the authors call the upper bands "supershift"? If I understood correctly, there is no antibody involved in this EMSA. Furthermore the authors state they quantified by qPCR relative to the input nucleosomes. How is this been done?

Minor issue:

5) How do the authors mechanistically explain DHT dependent but AR independent GATA2 action? How does the hormone influence GATA2 without involvement of AR?

Reviewer #2 (Remarks to the Author):

Critique:

This manuscript provides a tour de force analysis of "structural and mechanistic insight into the dynamics of pioneer factors governed nucleosome reorganization" in a prostate cancer cell line, LNCaP. The data are comprehensive and the experiments well-executed. The authors' approach is summarized in the discussion as: "1) we first identify the PF/TF-associated condensed 263 nucleosome states; 2) we identify the PF/TF binding borders within the nucleosomes; 3) we determine whether the PF/TF is accompanying the nucleosome switches under at least two biological conditions; and 4) we perform competitive nucleosome-binding assays to validate the pioneering capacity".

I found no major points of critique.

Minor points:

1. English edits are necessary through-out the manuscript. For example, the long sentence in the abstract (line22- line26) should be broken up (it has two semi-colons).
2. Data should be considered plural (singular = datum; abstract line27).
3. An important reference in the present context was omitted (PMID: 23034120).
4. It is stated in the Methods that "In brief, LNCaP cells were exposed to 10 nM DHT or vehicle (Veh) for 4 hrs". Was the vehicle ethanol, and if so at what concentration. It is known that ethanol has effects on cells.

Reviewer #3 (Remarks to the Author):

In the present study Li and co-authors generated a large multi-omics data set to explore the nucleosome landscape in LNCap cell model (androgen-sensitive human prostate adenocarcinoma cells) under vehicle and androgen-treated conditions. They define functional nucleosomal states based on histone marks, genomic regions, nucleosome positioning, spacing and regularity. They also observe dynamic switches among nucleosomal states upon androgen stimulation accompanied with differential binding of multiple pioneer transcription factors including Gata2. Furthermore, they show that Gata2 binds nucleosomal DNA and that, upon androgen stimulation, there is an increase in chromatin accessibility at its binding sites. Gata2, upon androgen stimulation, associates with Sox9 and Gata2 depletion reduces Sox9 occupancy at some Wnt/b-catenin signaling genes.

Overall, I have two main concerns about this study:

1. I do not see a novel noncanonical pioneering model for Gata2. It is already well known that pioneer factors a) bind nucleosomal DNA in vitro (Cirillo et al. Mol. Cell 2002 PMID 11864602; Soufi et al. Cell 2015 PMID 25892221) and in vivo (Meers et al. Mol. Cell 2019 PMID: 31253573). b) upon binding initiate a cascade of events leading to chromatin remodeling and recruitment of other transcription factors (Iwafuchi-Doi&Zaret, Genes&Dev, 2014 PMID: 25512556; Mayran&Drouin, JBC, 2018 PMID 29507097; Li et al. Genes&Dev, 2018, PMID 29440261; Sherwood et al. 2014 Nat. Biotechnol PMID: 24441470).
2. The manuscript misses an accurate description of the data (I will be more specific below) making hard sometimes to judge whether the conclusions are based on adequate experimental evidence.

Major points:

- 1) The functional states shown Fig.2E should be accompanied by visual examples (genome browser views)
- 2) The authors observe an increase of chromatin opening at Gata2 binding sites upon androgen stimulation. However, an important question remains open: Is Gata2 necessary for chromatin opening? The authors might address this question by using a Gata2-KO cell line.
- 3) The authors show that Gata2 depletion reduces Sox9 occupancy at 16 Wnt/b-catenin signaling genes (Fig6D). To support the claim that Gata2 recruits Sox9 it would be important to make this analysis genome-wide.
- 4) It would be important to show quality controls for the specificity of the Gata2 and Sox9 antibodies, in particular regarding other Gata/Sox factors eventually expressed in the LNCap cell system.
- 5) Fig6A: it would be important here to test also the supershift ability of Sox9. Fig6 C: inputs for vehicle and DHT conditions should be shown together with the levels of the bait

proteins. A Nuclease (like benzonase) treatment should be included to support the physical interaction. Fig.6D: claims of differences should be supported by an adequate statistical test.

6) Supplementary figure legends are missing.

7) Figures 1-5 show extensive data analysis of the large multi-omics data generated by the authors. However only "Detection of nucleosomes and defining nucleosomal states" are described in the methods. The computational analysis should be explained in detail and linked to each Figure. Codes should also be accessible.

8) Lines 200-201 "We further examined the open chromatin changes for both GATA2-associated nucleosome states switching by ATAC-seq data ". These data have not present in the GEO accession number associated to this manuscript. Are published data? Additionally Lines 211-214: "Remarkably, SOX9 was found as one of the top enriched motifsand identified as a major co-binding TF on WNT signaling genes by a publicly available database collected all ChIP-seq of TFs from ENCODE and ChEA". Also the data generated by others but analyzed in this manuscript should be properly documented with references and accession numbers.

Minor points:

9) Lines 120-122 "we compared the density of histone mark-enriched nucleosomes and gene expression, i.e. No. of nucleosomes per 1000 bp, in three regions, and found active marks, H3K4me1/2/3 and H3K27ac had higher density than repressive marks, H3K9me3 and H3K27me3 in all three regions ". Since the authors analyze mono-nucleosomes (see methods pag11) could this difference simply reflect the lower MNase accesibility of heterochromatic regions?

10) lines 177-178: "Furthermore, we found there were 257 unique GATA2 genes accompanying from S4 to RAS1 and 293 unique GATA2 genes accompanying from S3 to RAS2 respectively". Which criteria have been used for defining Gata2 genes?

11) Fig6a: how many replicates? What do the error bars show?

12) Fig6b: Which bioinformatic pipeline has been used to identify the differently expressed genes? Which thresholds?

13) Fig6d: how many replicates? What do the error bars show?

14) Methods, Chip-qPCR assay (pag13): what is the rationale of using GAPDH for normalization in a ChIP experiment?

16) In Fig.2b; Fig.3c,d; Fig.5d the color legend is missing

Response to Reviewers' Comments

Reviewer #1 (Remarks to the Author):

Comment 1. Major issues 1) Reporting/Description of what exactly has been done is missing in Materials and Methods and Figure Legends. For example, Figure 1b: Pearson correlation coefficient has been calculated on what metric? "# of % Aligned bases" is an unknown magnitude to me. Please explain better. Furthermore, 4 decimal places for r are pointless. Anyway, to claim reproducibility for some data it would be important to know how robust the biologically relevant signal is. In terms of nucleosomes this could be the robustness of dyad positions. Anyway, details about what kind of data has been processed how is missing in almost if not all legends.

Response: We have added more details for data processing steps in all Figure Legends from Figures 1-6 (see Pages 22-23) as well as in **Methods** (see Pages 13-14). Furthermore, we revised the axis labels of Figure 1b as "Reads of replicate 1/2 (x1000)" and the r coefficients. As suggested, a new figure panel **Figure 1c** was produced to show the robustness nucleosome positioning signals of Mono, Di-, Tri and Penta-nucleosomes.

Comment 2. Most if not all figures lack information on how many instances and/or replicates have been analyzed. e.g. how many promoters in Figure 2a, do we see here an average of replicate experiments? This information is crucial and missing in essentially all figure panels. It would be interesting to see in 4d the tracks for the different replicates.

Response: The information has been added in Figure Legends of all Figures 1-6. We have included a total of 21,319 promoters in genome-wide scale in **Figure 2a** and added the details of an average of replicates analyzed in the legend of **Figure 2** (see Page 22). A revised **Figure 4d** showed the tracks for both replicates in two different

conditions.

Comment 3. The description of the state classification method is completely cryptic. At which point and based on which metric and method does classification actually take place? It would be great to have a code repository not only for this method but most of the procedures applied in this study as it appears impossible to understand in detail what the authors did.

Response: The details of processing state classification and other bioinformatic analysis steps have been added in **Methods** (see Page 13). The codes for the analysis approaches have been submitted in Github as following: www.github.com/tianbao365/Nuc-PF.

Comment 4. Bioinformatic procedures such as mapping, nucleosome calling should be reported in detail with all parameters, version numbers of softwares as well as the versions of both genome and genome annotations used.

Response: The details for bioinformatic procedures have been added in **Methods** (see Pages 13-14), including the parameters, version of softwares, genome and genome annotations.

Comment 5. Figures are incomplete and/or contain unnecessary, unscientific elements. Fig 1c: 3D pie-chart with a mysterious arrow on the right side. 3D representations of pie-charts are unnecessary. The figure contains too much visual information for the simple message "50% of the genome is NDR". However, this information is problematic as this number appears way too high. I am actually concerned that technical reasons prevent the authors from calling nucleosomes on large parts of the genome.

Response: Original **Figure 1c** has been removed as suggested. We agreed with reviewer that there might be technical reasons to prevent us from calling all nucleosomes throughout the genome.

Comment 6. Fig 1d: why put an arrow (with shadow) and not a horizontal dashed line at 187 bp?

Response: Figure 1d has been revised as a horizontal dash line as suggested.

Comment 7. Axes labels are missing (e.g. Figs 2a, 2b 2c), legends are barely readable (e.g. Fig 2c), scales/ranges for color codes missing (Figs 3c, 3d)

Response: The axes labels are added in **Figs. 2a, 2b and 2c**. Legend of Fig. 2c has been revised to be much clearer. Color code scales have been added in **Figs. 3c and 3d**.

Comment 8. Fig 6d: given the skewed nature of ratios, the enrichment/fold changes in 6d should be displayed on log scales.

Response: The enrichment/fold changes in **Fig. 6d** was displayed on log scales.

Comment 9. What is the nature of error bars in Figs 2d/6a/6d?

Response: The details of error bars have been added in Figure Legends as following:

Fig. 2d: The error bar showed the standard deviation for an average number of nucleosomes with specific histone marks. **Fig. 6a** and **6d** showed the standard deviation for triplicates of qPCR results.

Comment 10. What is the ordering principle in the Fig 2b heatmaps? what are the

reference points for these heatmaps?

Response: The heatmap of **Fig. 2b** is sorted by the distribution of specific histone marks and the nucleosome dyads was set as 0 for each nucleosome. The column is the number away from nucleosome dyads by nucleobase and rows were sorted by the distance between the highest score and its nucleosome dyad. **Fig. 2b** illustrated the nucleosome spacing pattern under different histone marks.

Comment 11. Biological replicates. In none of the analyses it is clear how information of biological replicates has been incorporated. In fact, for some analyses e.g. state changes it would be very important to see how consistent the state changes are by having separate analyses for the (two?) replicates.

Response: We combined the replicates in original analyses process. As the reviewer's suggestion, we re-analyzed some panels in **Fig. 3** and **Fig. 4** and did a parallel visualization for two replicates in **Fig. 4d**.

Comment 12. The EMSA is an important experiment but without free DNA it is impossible to judge whether nucleosomes have been bound or not. please include a) a lane just with free DNA and lanes with free DNA and GATA2. Why do the authors call the upper bands "supershift"? If I understood correctly, there is no antibody involved in this EMSA. Furthermore, the authors state they quantified by qPCR relative to the input nucleosomes. How is this been done?

Response: The EMSA was perform with Native-PAGE system, the free DNA is small fragment and goes very fast in Native-PAGE than the nucleosomes or GATA2. We provided unbound free DNA with bounded nucleosomes as **Supplementary Fig. S17**. The supershift means the different binding models between nucleosome and GATA2 in vitro, which may change the migration rate in Native-PAGE. In addition, the reviewer is right that there is no antibody involved in the EMSA. We cut the

supershift band to perform qPCR according to the protocol of EMSA.

Comment 13. Minor issue: 5) How do the authors mechanistically explain DHT dependent but AR independent GATA2 action? How does the hormone influence GATA2 without involvement of AR?

Response: The classic genomic model for androgens presumes that androgens can freely enter the cytoplasm and bind to and activate specific intracellular ARs. The bound ARs act as transcription factors and bind as homodimers or heterodimers to specific ARE to regulate target genes. However, over past two decades numerous studies have concluded that androgen responses involve non-classical and initially non-genomic mechanisms where such actions can be mediated via multiple pathways such as through androgen membrane or membrane associated receptors/binding proteins, changes in membrane flexibility, changes in [Ca²⁺], activation of second messenger pathway or a combination of some or all of these mechanisms. We speculate that this noncanonical GATA2 pioneer function may be via this non-genomic mechanism. Future work may be focused on delineating the detailed pathways of androgen-mediated GATA2 transcription in prostate cancer progression.

Reviewer #2 (Remarks to the Author):

Comment 1. English edits are necessary through-out the manuscript. For example, the long sentence in the abstract (line22- line26) should be broken up (it has two semi-colons).

Response: We revised the long sentence in the abstract and did comprehensive English editing throughout the manuscript.

Comment 2. Data should be considered plural (singular = datum; abstract line27).

Response: The error has been corrected and we did proof-reading again for the whole manuscript.

Comment 3. An important reference in the present context was omitted (PMID: 23034120).

Response: Thanks for the review comment. We added the reference as Reference 24.

Comment 4. It is stated in the Methods that “In brief, LNCaP cells were exposed to 10 nM DHT or vehicle (Veh) for 4 hrs”. Was the vehicle ethanol, and if so at what concentration. It is known that ethanol has effects on cells.

Response: The vehicle condition is referred as DMSO, which is also solvent of DHT. We provided the details in the revised manuscript (See Page 11).

Reviewer #3 (Remarks to the Author):

Comment 1. 1. I do not see a novel noncanonical pioneering model for Gata2. It is already well known that pioneer factors a) bind nucleosomal DNA in vitro (Cirillo et al. Mol. Cell 2002 PMID 11864602; Soufi et al. Cell 2015 PMID 25892221) and in vivo (Meers et al. Mol. Cell 2019 PMID: 31253573). b) upon binding initiate a cascade of events leading to chromatin remodeling and recruitment of other transcription factors (Iwafuchi-Doi&Zaret, Genes&Dev, 2014 PMID: 25512556;

Mayran&Drouin, JBC, 2018 PMID 29507097; Li et al. Genes&Dev, 2018, PMID 29440261; Sherwood et al. 2014 Nat. Biotechnol PMID: 24441470).

Response: We are aware of these previous studies in pioneer factors fields and have cited some of these papers in this work. As we pointed it out in **Discussion** (Page 10), the PF functionality and capacity should be interpreted tightly with the specific biological context. What we illustrated the noncanonical pioneering model for Gata2 is in the context of hormone-induced prostate cancer. So far, all of previous work support an archetypical pioneer function of FOXA1 and GATA2 such that FOXA1 or GATA2 opens the condensed chromatin to mainly serve for an AR binding activity under DHT-treated condition (**Supplementary Figures S20-S21**, Robinson et al., *Oncogene*, 33:5666–5674 (2014); Jin et al., *Nature Comms*, 5:3972 (2014), Wu et al., *Nucleic Acids Res.*, 42:6 3607–3622 (2014); Bohm et al., *Oncogene*, 28:3847–3856 (2009)). However, our work clearly demonstrated that GATA2 pioneer action exerts in an AR-independent manner and regulates specific downstream signaling pathways (**Figs. 5b, c**). We speculate this noncanonical pioneer action of GATA2 may contribute to the progression to aggressive prostate cancer through a non-genomic mechanism. Future work may be focused on delineating the detailed pathways of androgen-mediated GATA2 transcription in prostate cancer progression.

Comment 2. The manuscript misses an accurate description of the data (I will be more specific below) making hard sometimes to judge whether the conclusions are based on adequate experimental evidence.

Response: We have added more accurate descriptions of the steps of generating and processing data. Point to point responses have been addressed to answer specific questions below and accordingly the main text has been revised (see Pages 13-14, 22-23).

Comment 3. Major points: 1) The functional states shown Fig.2E should be accompanied by visual examples (genome browser views)

Response: The genome browser views for functional states in Fig. 2E has been added as **Supplementary Fig. S3**.

Comment 4. 2) The authors observe an increase of chromatin opening at Gata2 binding sites upon androgen stimulation. However, an important question remains open: Is Gata2 necessary for chromatin opening? The authors might address this question by using a Gata2-KO cell line.

Response: In addition to the observation of an increase of chromatin opening at Gata2 binding sites upon androgen stimulation (**Fig. 5a**), we also conducted an Open Chromatin Assay on 20 Wnt/ β -catenin signaling genes and found 17 of 20 gene showed an increase of chromatin opening (see **Supplementary Fig. S18**). However, we failed to generate a GATA2-KO LNCaP subline due to its indispensable function in LNCaP cells. Therefore, we didn't conduct the open chromatin assay for GATA2-KO cells. In the following mechanistic study, we will perform RIME of GATA2 to identify other chromatin remodeling factors within the same complex of GATA2 and understand how they work together to open chromatin upon DHT treatment.

Comment 5. 3) The authors show that Gata2 depletion reduces Sox9 occupancy at 16 Wnt/ β -catenin signaling genes (Fig6D). To support the claim that Gata2 recruits Sox9 it would be important to make this analysis genome-wide.

Response: We did perform a genome-wide analysis for the association between GATA2 and SOX9 with ENCODE datasets and provided as **Supplementary Table S8**.

Comment 6. 4) It would be important to show quality controls for the specificity of the Gata2 and Sox9 antibodies, in particular regarding other Gata/Sox factors eventually expressed in the LNCap cell system.

Response: The antibodies we used were CHIP grade and could distinguish other Gata/Sox factors. GATA2 antibody (ab22849) could only detect GATA2 and not for GATA1 or GATA3; SOX9 antibody (ab3697) could separate SOX9 from other SOX family proteins. We also provided the test for the specificity of GATA2 and SOX9 antibodies in LNCaP cell systems as **Supplementary Fig. S19**.

Comment 7. 5) Fig6A: it would be important here to test also the supershift ability of Sox9. Fig6 C: inputs for vehicle and DHT conditions should be shown together with the levels of the bait proteins. A Nuclease (like benzonase) treatment should be included to support the physical interaction. Fig.6D: claims of differences should be supported by an adequate statistical test.

Response: The EMSA experiment could provide the direct binding between a pioneer factor and nucleosomes. SOX9 was not able to directly bind to the nucleosomes, therefore, no bands were showed in Native-PAGE experiment. We also provided a Figure for unbound nucleosomes as **Supplementary Figure S17** for supporting the physical interaction. For Figure 6c, we provided the figures for bait proteins as revised Figure 6c followed the suggestion from the reviewer. For Figure 6d, we added statistical test marks in the revised figures.

Comment 8. 6) Supplementary figure legends are missing.

Response: We have added more details for all Supplementary Figure legends in the revised version.

Comment 9. 7) Figures 1-5 show extensive data analysis of the large multi-omics data generated by the authors. However only “Detection of nucleosomes and defining nucleosomal states” are described in the methods. The computational analysis should be explained in detail and linked to each Figure. Codes should also be accessible.

Response: We have added the details for all steps for bioinformatics analysis in Methods (see Page 13) as well as in figure legends (see Pages 22-23). The codes of scripts are also provided in Github as following: www.github.com/tianbao365/Nuc-PF.

Comment 10. 8) Lines 200-201 “We further examined the open chromatin changes for both GATA2-associated nucleosome states switching by ATAC-seq data “. These data have not present in the GEO accession number associated to this manuscript. Are published data? Additionally Lines 211-214: “Remarkably, SOX9 was found as one of the top enriched motifsand identified as a major co-binding TF on WNT signaling genes by a publicly available database collected all ChIP-seq of TFs from ENCODE and ChEA”. Also the data generated by others but analyzed in this manuscript should be properly documented with references and accession numbers.

Response: GEO accession number associated with Line 200-201 (GSE105116) was provided in the revised manuscript in **Methods** (Page 14). The datasets accession number associated with Line 211-214 were collected and provided as **Supplementary Table S4**. We also double checked all the accession numbers of datasets and added references throughout the revised manuscript.

Comment 11. Minor points: 9) Lines 120-122 “we compared the density of histone mark-enriched nucleosomes and gene expression, i.e. No. of nucleosomes per 1000 bp, in three regions, and found active marks, H3K4me1/2/3 and H3K27ac had higher density than repressive marks, H3K9me3 and H3K27me3 in all three regions “. Since

the authors analyze mono-nucleosomes (see methods page 11) could this difference simply reflect the lower MNase accessibility of heterochromatic regions?

Response: The length variations between nucleosomes with different histone marks were also reported in the other previous studies (Refs 13 & 18). We detected an average of the number of nucleosomes in 1000bp for measuring the density of nucleosomes and the variant length of linked DNA. Our results showed nucleosome spacing were associated with specific histone marks in the genome-wide analysis.

Comment 12. 10) lines 177-178: “Furthermore, we found there were 257 unique GATA2 genes accompanying from S4 to RAS1 and 293 unique GATA2 genes accompanying from S3 to RAS2 respectively”. Which criteria have been used for defining Gata2 genes?

Response: The GATA2 genes were defined as the gene with both of GATA2 binding in ChIP-exo dataset and specific nucleosome states change. The unique GATA2 genes refer to the genes that only GATA2 binding were detected but other transcript factors binding were not detected. We made it much clearer in the revised **Results** (see Page 7).

Comment 13. 11) Fig6a: how many replicates? What do the error bars show?

Response: Triplicates were conducted in Figure 6a and the error bars shows the standard deviation for three qPCR results. We also added the details in the figure legend.

Comment 14. 12) Fig6b: Which bioinformatic pipeline has been used to identify the differently expressed genes? Which thresholds?

Response: The differentially expressed genes were performed by STAR,

HTseq-count and DESeq2 with thresholds of $\log_2(|\text{folder change}|) > 1$ and p-values < 0.05 . We revised the method for bioinformatics analysis and provided more details in the revised manuscript (see Page 14).

Comment 15. 13) Fig6d: how many replicates? What do the error bars show?

Response: Triplicates were provided in Figure 6d for qPCR results and the standard deviation for three qPCR results were showed as error bars.

Comment 16. 14) Methods, Chip-qPCR assay (pag13): what is the rationale of using GAPDH for normalization in a ChIP experiment?

Response: GAPDH was provided for normalizing the variation between multiple times of qPCR, not for ChIP experiments, we have revised the confused method in revised manuscript.

Comment 17. 16) In Fig.2b; Fig.3c,d; Fig.5d the color legend is missing

Response: The color legends of **Fig. 2b**, **Figs. 3c,d** and **Fig. 5d** have been added in the revised figures.

Reviewers' comments:

Reviewer #1 (Remarks to the Author):

The authors have extensively revised the paper, most of which I am satisfied with.

One major concern remains hardly addressed, which is the issue of robustness/reproducibility

1.) to me there is still very little (if any) evidence to claim robustness of the effects and phenomena described.

related to Comment 1:

Panel 1b: what is the nature of the data points in the scatter plot?

Panel 1c: if at all, this depiction shows intraexperimental robustness.

Anyway those figures do not shed any light on the overall robustness.

related to Comment 11:

I insist that performing the analyses in figs 4a to c separately for the replicate datasets would allow for assessing the robustness of the biological changes described.

minor point:

2.) I doubt that the code shared on github can be used by anyone without instructions how the different functions/scripts have to be run. Could the authors please provide documentation for a workflow example?

Reviewer #3 (Remarks to the Author):

In my opinion this manuscript still has several issues.

My major concern is that the authors still have not addressed if Gata2 is required for chromatin opening.

The authors states in the text that Gata2 re-configures inaccessible to accessible nucleosomes. The examples are as follows:

abstract lines 26-28: Upon androgen stimulation, GATA2 re-configures inaccessible to accessible nucleosome state and subsequently acts as a master regulator to recruit SOX9 to enhance oncogenic Wnt/ β -catenin signaling in an AR-independent manner.

Introduction lines 76-79: Under the DHT-treated condition, GATA2 reconfigures inaccessible to accessible nucleosome state and subsequently it acts as a master regulator to recruit SOX9 to enhance oncogenic Wnt/ β -catenin signaling in an AR- independent manner.

Results lines 215-220: Taken together, our data suggested a noncanonical pioneer model of GATA2 that it initially functions as a PF binding at the edge of a nucleosome in an inaccessible crowding array; under the DHT-treated condition, it reconfigures inaccessible to accessible nucleosome state; subsequently it acts as a master regulator to recruit SOX9 to enhance Wnt/ β -catenin signaling in an AR-independent manner.

These statements clearly assume a causal link between Gata2 binding and an increase in chromatin accessibility. However, this conclusion is based only on indirect findings, namely correlating the

presence of Gata2 with increased chromatin accessibility upon androgen stimulation (Figures 5A, S18). I suggested to address this question by using a Gata2-KO cell line. The authors could not generate a GATA2-KO cell line due to its indispensable function in LNCaP cells but they have in their hands a Gata2 knock-down system (Figure 6B,D) that might serve the scope as well.

This is an important question and the authors could not deliver the critical data needed to support the model. Ultimately, they need to provide adequate experimental evidence or reconsider their statements and model.

Other issues:

1) Genome browser views for functional states. This has been added as Supplementary Fig. S3. Why only for states S1 to S4? Further, three of the four examples are not convincing: a) signals of H3K27ac in panel-A,B and H3K4me1 in panel-C are at background levels b) state S3 (panel C) should be , according to Fig 2E, promoter/proximal. How far is the closest gene here? I suggest including some genome browser views in the main figure.

2) Omics data generated in this manuscript or by others should be properly documented with references and accession numbers. I can not find any documentation for the expression data of Fig.3D and fig S15

3) Codes has been deposited on Github but they should be linked to the corresponding Figure/plot.

4) Fig. 6C: A Nuclease (like Benzonase) treatment should be included to rule out that the interaction between Gata2 and Sox9 is mediated by a DNA/RNA moiety.

5) The authors performed a genome-wide analysis for the co-binding between GATA2 (in LNCaP cells, this study) and SOX9 (VCaP cell line, data set GSE76451 from ENCODE). Co-bound regions are shown in Table S8. However, the VCaP cell line carries a TMPRSS2-ERG gene fusion leading Sox9 over-expression, resulting in higher levels of Sox9 compared to LNCap cells (see fig S10 of the corresponding publication Ma et al. SOX9 drives WNT pathway activation in prostate cancer. J Clin Invest 2016.PMID: 27043282). Higher levels of Sox9 protein could result in a quite different binding pattern by increasing, for example, the occupancy at low-affinity binding sites and ultimately biasing the co-binding analysis.

6) Quality controls for the specificity of the Gata2 and Sox9 antibodies used in ChIP. The authors claim that GATA2 antibody (ab22849) could only detect GATA2 and not GATA1 or GATA3; SOX9 antibody (ab3697) could separate SOX9 from other SOX family proteins. Adequate experimental evidence is missing. Supplementary Fig S19 shows only a western blot for Gata2 and Sox9. For example, Gata3 and Gata2 have almost identical MW (49 and 51kDa respectively). How the authors can exclude a cross-reactivity?

7) Claims of differences in Fig.6d and Fig.S18 should be supported by an adequate statistical test. Asterisks for statistical significance has been added but there is no information about the statistical test used and to which samples it refers to.

8) Which criteria have been used for defining Gata2 genes? Still not clear within which distance from a Gata2 binding sites a gene is considered a Gata2 gene.

Response to Reviewers' Comments

Reviewer #1

Comment 1. *The authors have extensively revised the paper, most of which I am satisfied with.*

Response: We appreciated that the reviewer has satisfied with most of our efforts in improving the quality of our manuscript.

Comment 2. *to me there is still very little (if any) evidence to claim robustness of the effects and phenomena described.*

related to Comment 1: Panel 1b: what is the nature of the data points in the scatter plot? Panel 1c: if at all, this depiction shows intraexperimental robustness. Anyway those figures do not shed any light on the overall robustness.

Response: The data points in panel 1b were the count of MNase-seq reads within a 10Kb bin size along the whole genome and used for plotting a Pearson correlation between two biological replicates. It has now been removed. To show the robustness of two replicates of MNase-seq data at a nucleosome level, we now used a bin size of 200bp of the count of MNase reads to plot a Pearson correlation. The new **Fig. 1b** showed a representative Chromosome 10 with a higher r value ($=0.95$) and a very small p-value ($<2.2e-16$). We further showed the robust correlations of detected nucleosomes in each of 23 chromosomes between two replicates in the new **Fig. 1c**, with all r values larger than 0.93. We added the detailed description in the main text (Page 5).

Comment 3. *related to Comment 11: I insist that performing the analyses in figs 4a to c separately for the replicate datasets would allow for assessing the robustness of the biological changes described.*

Response: We performed the analyses for **Fig. 4a-e** separately for the replicate datasets in the revised manuscript.

Comment 4. *I doubt that the code shared on github can be used by anyone without*

instructions how the different functions/scripts have to be run. Could the authors please provide documentation for a workflow example?

Response: We provided a workflow and an instruction for detailing each step on github for guiding the users on how to perform the analysis.

Reviewer #3

Comment 1. *My major concern is that the authors still have not addressed if Gata2 is required for chromatin opening. The authors states in the text that Gata2 re-configures inaccessible to accessible nucleosomes. The examples are as follows:*

abstract lines 26-28: Upon androgen stimulation, GATA2 re-configures

Introduction lines 76-79: Under the DHT-treated condition, GATA2 reconfigures....

Results lines 215-220: Taken together, our data suggested a noncanonical pioneer model of GATA2 that

These statements clearly assume a causal link between Gata2 binding and an increase in chromatin accessibility. However, this conclusion is based only on indirect findings, namely correlating the presence of Gata2 with increased chromatin accessibility upon androgen stimulation (Figures 5A, S18).

Response: We concurred with this concern. Now we have performed ATAC-seq in LNCaP cells under the siControl vs siGATA2 conditions as the reviewer suggested to demonstrate that GATA2 is indeed involved in governing chromatin accessibility and nucleosome organization. Please see more in the response to Comment 2.

Comment 2. *I suggested to address this question by using a Gata2-KO cell line. The authors could not generate a GATA2-KO cell line due to its indispensable function in LNCaP cells but they have in their hands a Gata2 knock-down system (Figure 6B,D) that might serve the scope as well. This is an important question and the authors could not deliver the critical data needed to support the model. Ultimately, they need to provide adequate experimental evidence or reconsider their statements and model.*

Response: We performed ATAC-seq in LNCaP cells under the control and siGATA2 conditions. Our new results demonstrated that a majority of 413 GATA2-governed S4/3-RAS1/2 switching genes significantly reduced the chromatin accessibility and overall 29.6% open chromatin regions in a genome-wide scale were lost after knockdown GATA2 gene (**Fig. 6b** and **Supplementary Fig. S20**). Together, these results establish a causal link between GATA2 binding and chromatin accessibility and nucleosome reorganization. The new datasets were submitted in GEO, GSE182529. We revised the main text accordingly (Page 9).

***Comment 3.** Genome browser views for functional states. This has been added as Supplementary Fig. S3. Why only for states S1 to S4? Further, three of the four examples are not convincing: a) signals of H3K27ac in panel-A,B and H3K4me1 in panel-C are at background levels b) state S3 (panel C) should be, according to Fig 2E, promoter/proximal. How far is the closest gene here? I suggest including some genome browser views in the main figure.*

Response: We have now made genome browser views for all functional nucleosome states and provided it in the main figure, **Fig. 2f** and in a **Supplementary Fig. S3**. In our old version of visualization, we agreed with the reviewer that the signals of H3K27ac in Panel-A,B and H3K4me1 in panel -C were not considered as peak signals and already treated as background throughout our previous analysis process. The functional states were identified as the combination of nucleosome positioning level, spacing and histone modifications, thus states may not include all the histone modifications in some specific loci. Regarding how far is the closest gene, please see the response to Comment 10.

***Comment 4.** Omics data generated in this manuscript or by others should be properly documented with references and accession numbers. I can not find any documentation for the expression data of Fig.3d and Fig S15.*

Response: We provided the GEO access number for the datasets of **Fig. 3d** and **Fig. S15** in the revised manuscript. **Fig. 3d** applied the same datasets as **Fig. 3c** and

combined with our own data of nucleosome states. Datasets of ChIP-exo of AR (GSE143907) for **Fig. S15** and **S21** were provide in **Supplementary Table S4** and in main text (Page 16).

***Comment 5.** Codes has been deposited on Github but they should be linked to the corresponding Figure/plot.*

Response: We provided the link between the individual code and the corresponding figure. Further, we also provided a workflow and an instruction for users step by step on how to perform the analysis.

***Comment 6.** Fig. 6C: A Nuclease (like Benzonase) treatment should be included to rule out that the interaction between Gata2 and Sox9 is mediated by a DNA/RNA moiety.*

Response: After we re-submitted the revision, GATA2 antibody (ab22849) used for co-IP experiment has been discontinued by abcam. Therefore, in order to reveal the relationship between GATA2 and SOX9, we have now used validated GATA2 and SOX9 antibodies (see also Response to Comment 8) to perform new ChIP-seq of SOX9 in Veh and DHT-treated LNCaP cells at siControl and siGATA2 conditions, as well as new ChIP-seq of GATA2 in Veh and DHT-treated LNCaP cells at siControl and siSOX9 conditions (see also response to Comments 7). Our new analyses demonstrate genomic co-binding between GATA2 and SOX9 and establish the hierarchical chromatin binding relationship between these two transcription factors. We have added the results in new **Fig 6c** and **Supplementary File S5**. Together with **Fig 6a,b,d**, we elicited a novel noncanonical pioneer role of GATA2 that it initially functions as a PF binding at the edge of a nucleosome in an inaccessible crowding array, upon DHT stimulation, GATA2 re-configures inaccessible to accessible nucleosome state and subsequently acts as a master regulator to recruit SOX9 to enhance oncogenic Wnt/ β -catenin signaling.

***Comment 7.** The authors performed a genome-wide analysis for the co-binding between GATA2 (in LNCaP cells, this study) and SOX9 (VCaP cell line, data set GSE76451 from ENCODE). Co-bound regions are shown in Table S8. However, the*

VCaP cell line carries a TMPRSS2-ERG gene fusion leading Sox9 over-expression, resulting in higher levels of Sox9 compared to LNCaP cells (see fig S10 of the corresponding publication Ma et al. SOX9 drives WNT pathway activation in prostate cancer. J Clin Invest 2016.PMID: 27043282). Higher levels of Sox9 protein could result in a quite different binding pattern by increasing, for example, the occupancy at low-affinity binding sites and ultimately biasing the co-binding analysis.

Response: We appreciate that the reviewer raised this concern. We now generated four new ChIP-seq of SOX9 in Veh and DHT-treated LNCaP cells under siControl and siGATA2 conditions. The co-bound regions in both Veh and DHT conditions were provided as a new **Supplementary File S5**. Our results showed that GATA2 functions upstream of SOX9 and is able to recruit SOX9 in LNCaP cells upon DHT treatment (new **Fig. 6c**). The new datasets were submitted in GEO, GSE182529.

***Comment 8.** Quality controls for the specificity of the Gata2 and Sox9 antibodies used in ChIP. The authors claim that GATA2 antibody (ab22849) could only detect GATA2 and not GATA1 or GATA3; SOX9 antibody (ab3697) could separate SOX9 from other SOX family proteins. Adequate experimental evidence is missing. Supplementary Fig S19 shows only a western blot for Gata2 and Sox9. For example, Gata3 and Gata2 have almost identical MW (49 and 51kDa respectively). How the authors can exclude a cross-reactivity?*

Response: To address the concerns of antibody cross-reactivity, we have silenced GATA2 in LNCaP cells and examined protein expression of GATA2 and GATA3, and knocked down SOX9 in LNCaP cells to examine protein expression of SOX9 and SOX2. Compared with GATA2 and SOX9, GATA3 and SOX2 expressed at very low levels in LNCaP cells. Importantly, GATA2 (above 50 kd) antibody used in ChIP did not detect GATA3 (below 50 kd), and SOX9 (below 75 kd) antibody used in ChIP failed to detect SOX2 (35 kd). Please note that GATA2 antibody (ab22849) and SOX9 antibody (ab3697) have been discontinued by abcam, and we have used GATA2 antibody (sc-9008) and SOX9 antibody (AB5535) validated by cross-reactivity assays in our new ChIP-seq assays.

WB figures for source data

Comment 9. *Claims of differences in Fig.6d and Fig.S18 should be supported by an adequate statistical test. Asterisks for statistical significance has been added but there is no information about the statistical test used and to which samples it refers to.*

Response: A detailed statistical tests were provided in **Fig. 6d** and **Fig. S18**. The statistical analysis of qPCR results were performed by t-test and the asterisks present the p-value < 0.05.

Comment 10. *Which criteria have been used for defining Gata2 genes? Still not clear within which distance from a Gata2 binding sites a gene is considered a Gata2 gene.*

Response: The detailed description of which distance from a GATA2 binding sites to the TSS of a gene is considered a GATA2 gene were provided in **Methods** (Page 14). A GATA2 associated gene was defined as the closest gene of GATA2 border bindings and each GATA2 border pair was assigned to only one gene according to the order of following criterion: gene body region (TSS~ TES of a gene), promoter region (TSS~ -1Kb upstream of a gene), proximal region (-5Kb ~ -1Kb upstream of a gene), then distal region (-50Kb ~ -5Kb upstream of a gene) and no associated gene.

REVIEWER COMMENTS

Reviewer #1 (Remarks to the Author):

The authors have successfully addressed all my concerns. I now fully support publication of their manuscript.

Reviewer #3 (Remarks to the Author):

The authors have extensively revised the manuscript and satisfactorily addressed most of the concerns that I have raised but some points still need attention.

1) Causal link between Gata2 binding and chromatin accessibility: the authors perform ATACseq, in control and siGATA2 LNCaP cells and fig.6b shows that majority of 413 Gata2-governed S4/3-RAS1/2 switching genes have reduced chromatin accessibility. However, according to Gata2 ChIPseq performed by the authors (GEO GSE182529), Gata2 binds several thousands of loci in these cells and it would be important to extend the chromatin accessibility analysis to all Gata2 binding sites. This will reveal to which extent the causal link between Gata2 binding and chromatin accessibility takes place. Additionally less ATAC peaks in siGATA2 compared to control LNCaP cells (fig S20) is not an evidence of causality between Gata2 binding and chromatin accessibility since direct and indirect effects are not distinguishable.

2) Genome-wide analysis for the co-binding between Gata2 and Sox9: With the new generated ChIPseq data the authors detect in Veh and DHT-treated LNCaP cells respectively 746 and 974 co-bound regions (Supplementary files S5). In fig 6C they analyse deeper the co-binding enrichment, upon Sox9 or Gata2 depletion, at 278 Gata2-governed S4-RAS1 regions and see, for most of them, reduced Sox9 binding after Gata2 depletion but no changes in Gata2 binding after Sox9 depletion. It would be important to extend the analysis also to the other co-bound regions to show to which extent the hierarchical chromatin binding between these two transcription factors takes place in their experimental system. Furthermore, fig6C shows an appreciable heterogeneity in signal between the ChIPseq replicates that should be clarified.

3) Data deposited on GEO: in both GSE148935 (MNase-seq, MNase-ChIP-seq, ChP-ePENS of GATA2, ChIP-seq of H1) and GSE182529 (ATACseq, Gata2 and Sox9 ChIPseq) are missing the processed data files of replicates (peak bed files). In particular for GSE182529, in which SRA files are not accessible for review while status is private, it would be important to provide reviewers with appropriate processed files for data visualization (bedGraph/bigwig files).

Response to Reviewers' Comments

Reviewer #3 (Remarks to the Author):

Comment 1. “Causal link between *Gata2* binding and chromatin accessibility: ... it would be important to extend the chromatin accessibility analysis to all *Gata2* binding sites. This will reveal to which extent the causal link between *Gata2* binding and chromatin accessibility takes place. ... ”

Response: We have now extended the chromatin accessibility analysis for all GATA2 binding sites and added the result in **Suppl. Fig. S21**. The data showed GATA2-associated binding regions significantly reduced the chromatin accessibility, thus provided further evidence of causality between GATA2 binding and chromatin accessibility.

Comment 2. “Genome-wide analysis for the co-binding between *Gata2* and *Sox9*: ... It would be important to extend the analysis also to the other co-bound regions to show to which extent the hierarchical chromatin binding between these two

transcription factors takes place in their experimental system. Furthermore, fig6C shows an appreciable heterogeneity in signal between the ChIP-seq replicates that should be clarified.”

Response: We have now extended the analysis to all 746 and 974 co-bound regions of GATA2 and SOX9 and added a **Suppl. Fig. S22**. The heatmaps were conducted with all 1,155 co-binding sites, including 565 overlapped sites in both Veh and DHT-treated conditions, 181 unique in Veh condition and 409 unique in DHT-treated condition. The result showed that most of GATA2 and SOX9 co-bound regions reduced SOX9 binding after GATA2 depletion but no changes in GATA2 binding after SOX9 depletion under DHT-treated condition.

Regarding the heterogeneity in signal between the ChIP-seq replicates, we followed the guidance of ENCODE consortium requiring a minimum of two biological replicates in ChIP experiments, and made sure the Pearson correlation coefficient between two replicates was very higher. We then used MACS2 peak caller to identify GATA2 and SOX9 binding sites with a q value (minimum false discovery rate (FDR)) of 0.01, as well as nomodel and shift options in order to remove a technical bias and

peak shifting for ChIP-seq data. The co-binding sites of GATA2 and SOX9 binding were selected with a minimal of 10 bp overlapping peak regions and a maximal of 100 bp between two peaks' summit. However, we recognized that noise may be introduced during cell culture, many steps of ChIP, including technical issues in IP, library construction, or sequencing. Further, we used a z-score normalization to standardize the binding enrichment in building the heatmap (**Fig 6C**), which may enlarge the heterogeneity of two replicates.

***Comment 3.** "Data deposited on GEO: ... In particular for GSE182529, in which SRA files are not accessible for review while status is private, it would be important to provide reviewers with appropriate processed files for data visualization (bedGraph/bigwig files)."*

Response: We realized that the reviewer token 'alyfioecvhillcf' provided in our previous version may not allow GSE182529 SRA database access. We now updated it for the reviewer to access it. We also updated the GEO processed data files of two replicates independently for both of GSE148935 and GSE182529.

REVIEWER COMMENTS

Reviewer #3 (Remarks to the Author):

In my opinion, this manuscript still has unresolved issues regarding the Gata4-Sox9 co-binding.

1) The co-bound regions provided in Supplementary_File_S4 and Supplementary_File_S5 does not look to be supported by convincing enrichments of both TFs in the bigwig files finally appended to GEO GSE182529. In particular, I have checked DHT-treated samples: GSM5530925_LNCap cells_siCon_D_GATA2_rep1; GSM5530926_LNCap cells_siCon_D_GATA2_rep2; GSM5530927_LNCap cells_siCon_D_SOX9_rep1; GSM5530928_LNCap cells_siCon_D_SOX9_rep2.

From Supplementary_File_S4 (co-bound loci in DHT conditions) some examples are:

FLOT2 chr17 27230001 27230554
SHBG chr17 7518203 7518679
ELF1 chr13 41568207 41568434
MED1 chr17 37617546 37618183

From Supplementary_File_S5 (co-bound loci in WNT) some examples are:

WNT7B chr22 46409454 46409721
FZD4 chr11 86710943 86711682
PLCB2 chr15 40615457 40615870
WWTR1 chr3 149435555 149436325

2) The Gata2 and Sox9 ChIPseq data have a very high background and it is hard to distinguish a specific signal, especially for the latter.

3) Gata2 and Sox9 enrichments are shown only in separate heatmaps (Fig.6C and S22) from which it is not appreciable the co-binding. A heat map combining both Gata2 and Sox9 enrichments on the same sets of genomic regions should be provided.

In summary the co-binding of Gata2 and Sox9 is one of the key points of this manuscript and the authors could not deliver the critical data needed to support the model. I suggest either data refinement or removing the section of Gata2/Sox9.

Response to Reviewers' Comments

Reviewer #3 (Remarks to the Author):

In my opinion, this manuscript still has unresolved issues regarding the Gata4-Sox9 co-binding. "1) The co-bound regions provided in Supplementary_File_S4 and Supplementary_File_S5 does not look to be supported by convincing enrichments of both TFs in the bigwig files finally.." 2) "The Gata2 and Sox9 ChIPseq data have a very high background and it is hard to distinguish a specific signal, especially for the latter." 3) "Gata2 and Sox9 enrichments are shown only in separate heatmaps (Fig.6C and S22) from which it is not appreciable the co-binding. A heat map combining both Gata2 and Sox9 enrichments on the same sets of genomic regions should be provided."

In summary the co-binding of Gata2 and Sox9 is one of the key points of this manuscript and the authors could not deliver the critical data needed to support the model. I suggest either data refinement or removing the section of Gata2/Sox9.

Response: After carefully examining the co-binding regions of GATA2/SOX9, we agree with the reviewer that some of them didn't show higher enrichment of both TFs on the same loci. It could be the issues arose from the low expression of SOX9 in LNCaP cells, the specificity of SOX9 antibodies, the process in generation of ChIP-seq data or less stringency of peak identification. In order not to confuse the readers, we agreed with the reviewer and decided to remove this portion. We revised the main text and Fig. 6 accordingly, as well as removed Suppl. Fig. S22 and Suppl. Data Files S4-5. We should emphasize that the removal of this portion didn't impact any major conclusions of this study.